# Calibrated-Two Optional Randomized Response Techniques (C-TORRT) for the estimation of quantitative sensitive variable information

Mojeed Abiodun Yunusa[1], Ahmed Audu[1,2]*, Umar Usman[1], Kazeem Olalekan Aremu[3,4], Maggie Aphane[4]

1 Department of Statistics, Usmanu Danfodiyo University, Sokoto, Nigeria, 2 Department of Statistical Sciences, Sefako Makgatho Health Sciences University, Pretoria, South Africa, 3 Department of Mathematics, Usmanu Danfodiyo University, Sokoto, Nigeria, 4 Department of Maths and Applied Maths, Sefako Makgatho Health Sciences University, Pretoria, South Africa

* ahmed.audu@udusok.edu.ng

## Abstract

Accurate estimation of sensitive quantitative variables remains a challenge in survey research due to respondents' reluctance to disclose truthful information. While existing randomized response techniques (RRT) offer privacy protection, many suffer from inefficiencies and limited robustness. This study addresses this critical gap by proposing new classes of *Calibrated-Two Optional Randomized Response Techniques (C-TORRT)*, developed through calibration methods that incorporate auxiliary information to enhance estimation accuracy and respondent privacy. The theoretical framework of the proposed models demonstrates unbiasedness, reduced variance, higher privacy protection, and a superior combined metric of efficiency and privacy. Empirical studies based on real-life and simulated data showed that the proposed C-TORRT models consistently outperformed existing RRT models. For instance, under Population I, the proposed model achieved a variance of 21.76704, privacy level of 222.4369, and a percentage relative efficiency (PRE) of 608.93, compared to the Azeem et al. model with variance 142.4927 and PRE 93.02. Similarly, under Population II, the C-TORRT model reduced the variance to 4.3098 and raised the PRE to 499.60, a significant improvement over Gjestvang and Singh's variance of 21.5316 and PRE of 100. Real-life data application using academic records confirmed these findings, where the C-TORRT estimators yielded lower variance (0.8084), higher privacy levels (817.28), and smaller combined efficiency-privacy metrics (0.000991) compared to existing models. These results underscore the superior efficiency, precision, and privacy protection of the proposed C-TORRT models, making them robust alternatives for sensitive quantitative data collection.

**Data availability statement:** All relevant data are within the manuscript and its Supporting information files.

**Funding:** The author(s) received no specific funding for this work.

**Competing interests:** The authors have declared that no competing interests exist.

## Introduction

Reliable data on sensitive issues such as cases of rape, use of hard drug, illegal earning, sexual harassment, cheating in examination and other sensitive issues are not easily obtainable from respondents using direct method of questioning in a sensitive survey. In a sensitive survey, respondents would rather choose to provide untrue response or refuse to answer the sensitive question, because of fear of being stigmatized or punished by the law. To obtain more reliable information from the respondents [1], came up with an ingenious approach for data collection known as the randomized response technique (RRT). This method guarantees the privacy of the respondents and also conceals their responses. The technique was specifically designed for qualitative variable but was extended by [2] to quantitative variable [3] reintroduced this technique as additive model. Randomized response techniques (RRT) are valuable techniques used in surveys or questionnaires to collect sensitive data while maintaining the privacy of participants. Social desirability bias is minimized using the randomized response technique (RRT) and it ensures participants privacy in sharing sensitive information without any fear. RRT increases the chance that respondents will answer sensitive questions honestly, the data collected is generally more accurate. This is particularly useful in cases where researchers are studying controversial or sensitive issues. By guaranteeing confidentiality, RRT can increase participation rates in surveys or studies involving sensitive topics. The primary goal of RRT is to protect individual privacy by randomizing responses and ensuring that no one can determine the true response of any specific participant. This provides an extra layer of protection for individuals participating in surveys. RRT can be applied to various types of data collection, including face-to-face interviews, self-administered questionnaires, and online surveys, making it a versatile tool for researchers. By reducing the potential discomfort associated with divulging sensitive information, RRT can lead to lower non-response rates, resulting in more comprehensive and valuable data.

RRT application in different field of endeavors like health, education, criminology, policy making etc. cannot be overemphasized. Under health, It can be used to gather accurate data about sensitive health behaviors like substance abuse, sexual practices, and mental health issues. RRTs can help estimate the prevalence of various diseases, especially those that carries a significant social stigma (such as sexually transmitted infections or mental health disorders) or those related to sensitive behaviors (such as HIV or Hepatitis C). RRT can be used in survey to investigate how patients are using their medication. RRT approach helps researchers and policymakers to have better understanding of disparities in healthcare access, quality, and outcomes, particularly among disadvantaged groups. They can also be employed to assess the accessibility and utilization of healthcare services, particularly among vulnerable populations. This can provide insights into barriers to care and inform targeted interventions. Several authors have proposed different RRT models and estimators for estimation of population parameters of sensitive qualitative/quantitative variables. These authors include [4–22].

Auxiliary information refers to additional data that is related to the variable of interest and can be used to improve estimators' performance in sample surveys. Auxiliary

information can help reduce the variance of estimators, leading to more precise estimates [23]. This is particularly useful when dealing with small sample sizes or when estimating small subpopulations. Auxiliary information can be used to adjust for potential biases in the sample, such as non-response bias or under-coverage bias, resulting in more accurate estimates. By incorporating auxiliary information, estimators can become more efficient, requiring smaller sample sizes to achieve the same level of precision [24]. This can help reduce the costs associated with data collection. Auxiliary information can be utilized to handle missing data in surveys, either through imputation or by informing the choice of weights in estimation methods. Auxiliary information can come from various sources, such as administrative data, census data, or previous surveys [25]. This allows researchers to leverage existing data to improve estimators and enrich their analyses. Using auxiliary information to align estimates with known population benchmarks or external data sources can facilitate better comparisons across different surveys or time periods. One of the approaches in which the auxiliary information can be incorporated into models/estimators is the use of calibration techniques [26].

Calibration of estimators in sample surveys is a method used to adjust for discrepancies between the survey sample and the target population, particularly when the sample may not be fully a representative. Calibration adjusts estimators to align with known population totals or benchmarks, reducing bias and improving the accuracy of the estimates. Calibration helps reduce the impact of sampling variability, which can arise when a sample is not entirely a representative of the target population. Calibration can help correct for non-sampling errors such as frame coverage errors, non-response bias, and measurement errors. By aligning estimates to known population benchmarks, calibration ensures that estimates from different surveys or time periods can be compared more accurately. Calibrated estimators are generally more robust, as they account for uncertainties in the sampling process and reduce sensitivity to model assumptions. Calibration can incorporate auxiliary information from other sources, such as administrative data or census data, to further refine and improve the estimates. Recently, [27] and [28] developed calibrated estimators for sensitive variables under simple and stratified random sampling respectively. In the present, we aimed at proposing randomized response calibrated estimators for estimation of mean of sensitive study variable under simple random sampling schemes.

### Some existing quantitative randomized response schemes and their estimators

Let Y be a sensitive study variable which correlated with an auxiliary variable X from a population consists of N elements from which a sample of size n elements is drawn. Let S and T be scrambling variables which are uncorrelated with Y whose mean and variances are assumed to be known. Let Z be the scrambling response of Y. Then, the following notations are defined.

$E(Y_i) = \mu_y$ : Population mean of the sensitive study variable Y

$Var(Y_i) = \sigma_y^2$: Population variance of the sensitive study variable Y

$E(S) = \mu_s = \theta$: Population mean of the scrambling variable S

$Var(S) = \sigma_s^2$: Population variance of the scrambling variable S

$E(T) = \mu_T$: Population mean of the scrambling variable T

$Var(T) = \sigma_T^2$: Population variance of the scrambling variable T

$\bar{X} = N^{-1}\sum_{i=1}^{N} X_i$: Population mean of the auxiliary variable X

$\sigma_x^2 = N^{-1}\sum_{i=1}^{N}\left(x_i - \bar{X}\right)^2$: Population variance of the auxiliary variable

$\bar{x} = n^{-1}\sum_{i=1}^{n} x_i$: Sample mean of the auxiliary variable X

$s_x^2 = (n-1)^{-1}\sum_{i=1}^{n}\left(x_i - \bar{x}\right)^2$: Sample variance of the auxiliary variable

Some relevant and related reviewed literatures are presented as follows.

[11] proposed RRT model for estimating sensitive information as in Eq. (1).

$$Z_{(GS)i} = \begin{cases} Y_i + \alpha S & \text{with prob. } P = \frac{\beta}{\alpha+\beta} \\ Y_i - \beta S & \text{with prob. } 1 - P = \frac{\alpha}{\alpha+\beta} \end{cases}$$

(1)

where $\alpha$ and $\beta$ are constants determined by the interviewer.

The estimator of population mean $\mu_y$ and its variance using [11] model under the assumption that $E(S) = \mu_s \neq 0$ are given as in Eqs. (2) and (3) respectively and the combined metric of privacy level and efficiency denoted by $\delta_{GS}$ is given as in Eq. (4).

$$\hat{\mu}_{GS} = \frac{1}{n} \sum_{i=1}^{n} Z_i$$

(2)

$$Var(\hat{\mu}_{GS}) = \frac{1}{n} \left( \alpha\beta \left( \sigma_s^2 + \mu_s^2 \right) + \sigma_y^2 \right)$$

(3)

$$\delta_{GS} = \frac{\sigma_y^2 + \alpha\beta \left( \sigma_s^2 + \mu_s^2 \right)}{n\alpha\beta \left( \sigma_s^2 + \mu_s^2 \right)}$$

(4)

[22] proposed RRT model with one scramble variable for estimating sensitive information as in Eq. (5).

$$Z_{(AZ)i} = \begin{cases} Y_i + \alpha S, & P = \beta/(\alpha+\beta) \\ Y_i - \beta S, & 1 - P = \alpha/(\alpha+\beta) \end{cases}$$

(5)

where $\alpha$ and $\beta$ are constants determined by the interviewer.

The estimator of population means $\mu_y$ and its variance using [22] model under the assumption that $E(S) = 0$ are given as in Eqs. (6) and (7) respectively and the combined metric of privacy level and efficiency denoted by $\delta_{AZ}$ is given as in Eq. (8).

$$\hat{\mu}_{AZ} = \frac{1}{n} \sum_{i=1}^{n} Z_i$$

(6)

$$Var(\hat{\mu}_{AZ}) = \frac{1}{n} \left( \alpha\beta\sigma_s^2 + \sigma_y^2 \right)$$

(7)

$$\delta_{AZ} = \frac{\sigma_y^2 + \alpha\beta\sigma_s^2}{n\alpha\beta\sigma_s^2}$$

(8)

## Materials and methods

### Proposed calibration estimators

Let $G_1$ and $G_2$ be the sets of respondents belonging to the first and second categories of Z respectively in all the RRT models stated in Eqs. (1) and (5) having elements $n_1$ and $n_2$. Then, the models and their estimators can be generally written in the form defined Eqs. (9) and (10) respectively.

$$Z_i^{(kj)} = \begin{cases} \theta(y)_{1i} & \text{with probability } p^*, & i \in G_1 \\ \theta(y)_{2i} & \text{with probability } 1 - p^*, & i \in G_2 \end{cases} \tag{9}$$

$$\hat{\mu}^{(kj)} = \sum_{i \in G_1} W_1 \theta(y)_{1i} + \sum_{i \in G_2} W_2 \theta(y)_{2i} \tag{10}$$

where $W_1 = W_2 = \frac{1}{n}$, $\theta(y)_{1i} = Y_i + \alpha S$, $\theta(y)_{2i} = Y_i - \beta S$, $p^* = \beta / (\alpha + \beta)$

Motivated by [26], this study proposed two (2) calibration Schemes and Estimators to obtain two new classes of RRT models for sensitive variables,

**First proposed calibration estimator.** The first proposed calibration scheme and estimator of population mean is defined as in Eqs. (11) and (12) respectively

$$\hat{\mu}^{(1j)} = \sum_{i \in G_1} W_{11i} \theta(y)_{1i} + \sum_{i \in G_2} W_{12i} \theta(y)_{2i} \tag{11}$$

where $W_{11i}$ and $W_{12i}$ are the new calibration weights to be obtained by minimizing the chi-square distance $\Phi_1$ defined as in Equation (12).

$$\left. \begin{aligned} \min \ \Phi_1 &= \sum_{i \in G_1} \frac{(nW_{11i} - 1)^2}{2n\phi_{11i}} + \sum_{i \in G_2} \frac{(nW_{12i} - 1)^2}{2n\phi_{12i}} \\ s.t. \quad & \sum_{i \in G_1} W_{11i} x_{1i} + \sum_{i \in G_2} W_{12i} x_{2i} = \mu_x \end{aligned} \right\} \tag{12}$$

To compute new calibrated weights $W_{11i}$ and $W_{12i}$, Lagrange function $L_1$ is defined as in Eq. (13).

$$L_1 = \sum_{i \in G_1} \frac{(nW_{11i} - 1)^2}{2n\phi_{11i}} + \sum_{i \in G_2} \frac{(nW_{12i} - 1)^2}{2n\phi_{12i}} - \lambda \left( \sum_{i \in G_1} W_{11i} x_{1i} + \sum_{i \in G_2} W_{12i} x_{2i} - \mu_x \right) \tag{13}$$

Partially differentiating Eq. (13) with respect to $W_{11i}$, $W_{12i}$, and $\lambda$, equate the results to zero, we obtained Eqs. (14)–(16)

$$W_{11i} = n^{-1} (1 + \lambda \phi_{11i} x_{1i}) \tag{14}$$

$$W_{12i} = n^{-1} (1 + \lambda \phi_{12i} x_{2i}) \tag{15}$$

$$\sum_{i \in G_1} W_{11i} x_{1i} + \sum_{i \in G_2} W_{12i} x_{2i} = \mu_x \tag{16}$$

By substituting Eqs. (14) and (15) into Eq. (16) and solve for $\lambda$, Eq. (17) is obtained.

$$\lambda = \left( n\mu_x - \left( \sum_{i \in G_1} x_{1i} + \sum_{i \in G_2} x_{2i} \right) \right) \left( \sum_{i \in G_1} \phi_{11i} x_{1i}^2 + \sum_{i \in G_2} \phi_{12i} x_{2i}^2 \right)^{-1} \tag{17}$$

Substituting the value of $\lambda$ into Eqs. (14) and (15), the expressions for $W_{11i}$ and $W_{12i}$ are obtained as in Eqs. (18) and (19) respectively.

$$W_{11i} = \frac{1}{n} + \phi_{11i} x_{1i} \left( \mu_x - n^{-1} \left( \sum_{i \in G_1} x_{1i} + \sum_{i \in G_2} x_{2i} \right) \right) \left( \sum_{i \in G_1} \phi_{11i} x_{1i}^2 + \sum_{i \in G_2} \phi_{12i} x_{2i}^2 \right)^{-1} \tag{18}$$

$$W_{12i} = \frac{1}{n} + \phi_{12i}x_{2i}\left(\mu_x - n^{-1}\left(\sum_{i\in G_1} x_{1i} + \sum_{i\in G_2} x_{2i}\right)\right)\left(\sum_{i\in G_1}\phi_{11i}x_{2i}^2 + \sum_{i\in G_2}\phi_{12i}x_{2i}^2\right)^{-1} \tag{19}$$

Substituting Eqs. (18) and (19) in Eq. (11) to obtain the proposed calibration estimator as in Eq. (20)

$$\hat{\mu}^{(1j)} = \hat{\mu} + \hat{\beta}_{1j}\left(\mu_x - \bar{x}\right) \tag{20}$$

where $\bar{x} = n^{-1}\left(\sum_{i\in G_1} x_{1i} + \sum_{i\in G_2} x_{2i}\right)$, $\hat{\beta}_{1j} = \frac{\sum_{i\in G_1}\phi_{11i}x_{1i}\theta(y)_{1i} + \sum_{i\in G_2}\phi_{12i}x_{2i}\theta(y)_{2i}}{\sum_{i\in G_1}\phi_{11i}x_{1i}^2 + \sum_{i\in G_2}\phi_{12i}x_{2i}^2}$.

**Case 1:** Setting $\phi_{11i} = \phi_{12i} = 1$ in Eq. (20), member of $\hat{\mu}^{(1j)}$ denoted by $\hat{\mu}^{(11)}$ is obtained as in Eq. (21)

$$\hat{\mu}^{(11)} = \hat{\mu} + \hat{\beta}_{11}\left(\mu_x - \bar{x}\right) \tag{21}$$

where $\hat{\beta}_{11} = \frac{\sum_{i\in G_1} x_{1i}\theta(y)_{1i} + \sum_{i\in G_2} x_{2i}\theta(y)_{2i}}{\sum_{i\in G_1} x_{1i}^2 + \sum_{i\in G_2} x_{2i}^2}$

**Case 2:** Setting $\phi_{11i} = x_{1i}^{-1}$, $\phi_{12i} = x_{2i}^{-1}$ in Eq. (20), member of $\hat{\mu}^{(1j)}$ denoted by $\hat{\mu}^{(12)}$ is obtained as in Eq. (22)

$$\hat{\mu}^{(12)} = \hat{\mu} + \hat{\beta}_{12}\left(\mu_x - \bar{x}\right) \tag{22}$$

where $\hat{\beta}_{12} = \frac{\sum_{i\in G_1}\theta(y)_{1i} + \sum_{i\in G_2}\theta(y)_{2i}}{\sum_{i\in G_1} x_{1i} + \sum_{i\in G_2} x_{2i}} = \frac{\bar{z}^*}{\bar{x}}$.

The resultant estimator $\hat{\mu}^{(1j)}$ obtained in Eq. (20) can be expressed as in Eq. (23)

$$\hat{\mu}^{(1j)} = \sum_{i\in G_1} W_1\left\{\theta(y)_{1i} + \hat{\beta}_{1j}\left(\mu_x - x_{1i}\right)\right\} + \sum_{i\in G_2} W_2\left\{\theta(y)_{2i} + \hat{\beta}_{1j}\left(\mu_x - x_{2i}\right)\right\} \tag{23}$$

Compared Eq. (23) with Eq. (20), the first proposed modified RRT model is obtained as in Eq. (24).

$$Z^{(1j)} = \begin{cases} \theta(y)_1 + \hat{\beta}_{1j}\left(\mu_x - X_1\right) & \text{with prob. } p^* \\ \theta(y)_2 + \hat{\beta}_{1j}\left(\mu_x - X_2\right) & \text{with prob. } 1 - p^* \end{cases} \tag{24}$$

**Second proposed calibration estimator.** The second proposed calibration scheme and estimator of population mean is defined as in Eqs. (25) and (26) respectively.

$$\hat{\mu}^{(2j)} = \sum_{i\in G_1} W_{21i}\theta(y)_{1i} + \sum_{i\in G_2} W_{22i}\theta(y)_{2i} \tag{25}$$

where $W_{21i}$ and $W_{22i}$ are the new calibration weights to be obtained by minimizing the chi-square distance $\Phi_2$ defined as in Equation (26).

$$\left.\begin{array}{ll} \min\ \Phi_2 = \sum_{i\in G_1}\frac{(nW_{21i}-1)^2}{2n\phi_{21i}} + \sum_{i\in G_2}\frac{(nW_{22i}-1)^2}{2n\phi_{22i}} \\ s.t. \qquad \sum_{i\in G_1} W_{21i}x_{1i} + \sum_{i\in G_2} W_{22i}x_{2i} = \mu_x \\ \qquad \sum_{i\in G_1} W_{21i} + \sum_{i\in G_2} W_{22i} = \sum_{i\in G_1} W_1 + \sum_{i\in G_2} W_2 \end{array}\right\} \tag{26}$$

To compute new calibrated weights $W_{21i}$ and $W_{22i}$, Lagrange multiplier function $L_2$ is defined as in Eq. (27)

$$L_2 = \sum_{i \in G_1} \frac{(nW_{21i}-1)^2}{2n\phi_{21i}} + \sum_{i \in G_2} \frac{(nW_{22i}-1)^2}{2n\phi_{22i}} - \lambda_1 \left( \sum_{i \in G_1} W_{21i}x_{1i} + \sum_{i \in G_2} W_{22i}x_{2i} - \mu_x \right)$$
$$- \lambda_2 \left( \sum_{i \in G_1} W_{21i} + \sum_{i \in G_2} W_{22i} - \sum_{i \in G_1} W_1 - \sum_{i \in G_2} W_2 \right) \tag{27}$$

Partially differentiating (27) with respect to $W_{21i}$, $W_{22i}$, $\lambda_1$ and $\lambda_2$, equate the results to zero, Eqs. (28)–(31) are obtained respectively

$$W_{21i} = n^{-1} \left( 1 + \lambda_1 \phi_{21i} x_{1i} + \lambda_2 \phi_{21i} \right) \tag{28}$$

$$W_{22i} = n^{-1} \left( 1 + \lambda_1 \phi_{22i} x_{2i} + \lambda_2 \phi_{22i} \right) \tag{29}$$

$$\sum_{i \in G_1} W_{21i} x_{1i} + \sum_{i \in G_2} W_{22i} x_{2i} = \mu_x \tag{30}$$

$$\sum_{i \in G_1} W_{21i} + \sum_{i \in G_2} W_{22i} = \sum_{i \in G_1} W_1 + \sum_{i \in G_2} W_2 \tag{31}$$

By substituting Eqs. (28) and (29) into Eqs. (30) and (31), system of linear equation in Eq. (32) is obtained

$$\begin{pmatrix} T_1 & T_2 \\ T_2 & T_3 \end{pmatrix} \begin{pmatrix} \lambda_1 \\ \lambda_2 \end{pmatrix} = \begin{pmatrix} T_4 \\ 0 \end{pmatrix} \tag{32}$$

where $T_1 = \frac{1}{n} \left( \sum_{i \in G_1} \phi_{21i} x_{1i}^2 + \sum_{i \in G_2} \phi_{22i} x_{2i}^2 \right)$, $T_2 = \frac{1}{n} \left( \sum_{i \in G_1} \phi_{21i} x_{1i} + \sum_{i \in G_2} \phi_{22i} x_{2i} \right)$, $T_3 = \frac{1}{n} \left( \sum_{i \in G_1} \phi_{21i} + \sum_{i \in G_2} \phi_{22i} \right)$,

$T_4 = \mu_x - \frac{1}{n} \left( \sum_{i \in G_1} x_{1i} + \sum_{i \in G_2} x_{2i} \right)$.

By solving Eq. (32) for $\lambda_1$ and $\lambda_2$, Eq. (33) is obtained.

$$\lambda_1 = \frac{T_3 T_4}{T_1 T_3 - T_2^2}, \quad \lambda_2 = \frac{-T_2 T_4}{T_1 T_3 - T_2^2} \tag{33}$$

Substituting the values of $\lambda_1$ and $\lambda_2$ into Eqs. (28) and (29), the expressions for $W_{21i}$ and $W_{22i}$ are obtained as in Eqs. (34) and (35) respectively.

$$W_{21i} = n^{-1} \left( 1 + \frac{T_3 T_4}{T_1 T_3 - T_2^2} \phi_{21i} \theta(x)_{1i} - \frac{T_2 T_4}{T_1 T_3 - T_2^2} \phi_{21i} \right) \tag{34}$$

$$W_{22i} = n^{-1} \left( 1 + \frac{T_3 T_4}{T_1 T_3 - T_2^2} \phi_{22i} \theta(x)_{2i} - \frac{T_2 T_4}{T_1 T_3 - T_2^2} \phi_{22i} \right) \tag{35}$$

By substituting Eqs. (34) and (35) in Eq. (25), the proposed calibration estimator is obtained as in Eq. (36).

$$\hat{\mu}^{(2j)} = \hat{\mu} + \hat{\beta}_{2j} \left( \mu_x - \bar{x} \right) \tag{36}$$

where $\hat{\beta}_2 = \frac{T_3 T_5 - T_2 T_6}{T_1 T_3 - T_2^2}$, $T_5 = \frac{1}{n}\left(\sum_{i \in G_1} \phi_{21i} x_{1i} \theta(y)_{1i} + \sum_{i \in G_2} \phi_{22i} x_{2i} \theta(y)_{2i}\right)$, $T_6 = \frac{1}{n}\left(\sum_{i \in G_1} \phi_{21i} \theta(y)_{1i} + \sum_{i \in G_2} \phi_{22i} \theta(y)_{2i}\right)$.

**Case 1:** Setting $\phi_{21i} = \phi_{22i} = 1$ in Eq. (36), member of $\hat{\mu}^{(2j)}$ denoted by $\hat{\mu}^{(21)}$ is obtained as in Eq. (37)

$$\hat{\mu}^{(21)} = \hat{\mu} + \hat{\beta}_{21}\left(\mu_x - \bar{x}\right) \tag{37}$$

where $\hat{\beta}_{21} = \dfrac{\sum_{i \in G_1} x_{1i}\theta(y)_{1i} + \sum_{i \in G_2} x_{2i}\theta(y)_{2i} - \bar{x}\left(\sum_{i \in G_1}\theta(y)_{1i} + \sum_{i \in G_2}\theta(y)_{2i}\right)}{\sum_{i \in G_1} x_{1i}^2 + \sum_{i \in G_2} x_{2i}^2 - n\bar{x}^2}$

**Case 2:** Setting $\phi_{21i} = x_{1i}^{-1}$, $\phi_{22i} = x_{2i}^{-1}$ in Eq. (36), member of $\hat{\mu}^{(2j)}$ denoted by $\hat{\mu}^{(22)}$ is obtained as in Eq. (38).

$$\hat{\mu}^{(22)} = \hat{\mu} + \hat{\beta}_{22}\left(\mu_x - \bar{x}\right) \tag{38}$$

where $\hat{\beta}_{22} = \dfrac{\hat{\mu}_{Z^*}\left(\sum_{i \in G_1}\frac{1}{x_{1i}} + \sum_{i \in G_2}\frac{1}{x_{2i}}\right) - \left(\sum_{i \in G_1}\frac{x_{1i}}{\theta(y)_{1i}} + \sum_{i \in G_2}\frac{x_{2i}}{\theta(y)_{2i}}\right)}{\bar{x}\left(\sum_{i \in G_1}\frac{1}{x_{1i}} + \sum_{i \in G_2}\frac{1}{x_{2i}}\right) - 1}$,

The resultant estimator $\hat{\mu}^{(2j)}$ obtained in Eq. (36) can be expressed as in Eq. (39).

$$\hat{\mu}^{(2j)} = \sum_{i \in G_1} W_1 \left\{\theta(y)_{1i} + \hat{\beta}_{2j}\left(\mu_x - x_{1i}\right)\right\} + \sum_{i \in G_2} W_2 \left\{\theta(y)_{2i} + \hat{\beta}_{2j}\left(\mu_x - x_{2i}\right)\right\} \tag{39}$$

Compare Eq. (39) with Eq. (10), the second proposed modified RRT model (C-RRT-2) is obtained as in Eq. (40) (Table 1).

$$Z^{(2j)} = \begin{cases} \theta(y)_1 + \hat{\beta}_{2j}\left(\mu_x - X_1\right) & \textit{with prob. } p^* \\ \theta(y)_2 + \hat{\beta}_{2j}\left(\mu_x - X_2\right) & \textit{with prob. } 1 - p^* \end{cases} \tag{40}$$

## Properties of the proposed calibration RRT estimators

This subsection presents the theoretical properties (Expectation, variance, privacy level) of the proposed calibrated estimators.

**Theorem 1:** Given that $\hat{\beta}_{kj}$, $k, j = 1, 2$ is unbiased of $\beta_{kj}$, $k, j = 1, 2$, then, $E\left(\hat{\mu}^{(kj)}\right) = \mu_y$. That is, the estimators $\hat{\mu}^{(kj)}$, $k, j = 1, 2$ of population mean $\mu_y$ are unbiased.

**Proof:**

As $n \to N$, $\lim_{n \to N}\left(\hat{\beta}_{kj}\right) = \beta_{ij}$, $k, j = 1, 2$, where $\beta_{11} = \frac{\sum_{i=1}^{N} Y_i X_i}{\sum_{i=1}^{N} X_i^2} = \frac{\rho_{yx}\sigma_y\sigma_x + \mu_y\mu_x}{\sigma_x^2 + \mu_x^2}$, $\beta_{12} = \frac{\sum_{i=1}^{N} Y_i}{\sum_{i=1}^{N} X_i} = \frac{\mu_y}{\mu_x}$, $\beta_{21} = \frac{\rho_{yx}\sigma_y}{\sigma_x}$, $\beta_{22} = \frac{\mu_y\mu_{1/x} - \mu_{x/y}}{\mu_x\mu_{1/x} - N^{-1}}$.

**Table 1. Members of the proposed calibrated optional quantitative RRT model $Z_i^{(j)}$, $j = 1, 2$.**

| Existing RRT Models | Corresponding Members of $Z^{(kj)}$, $k, j = 1, 2$ |
|---|---|
| Gjestvang and Singh (2009) $Z_{(GS)i} = \begin{cases} Y_i + \alpha S, & p^* = \frac{\beta}{\alpha+\beta} \\ Y_i - \beta S, & 1 - p^* = \frac{\alpha}{\alpha+\beta} \end{cases}$ | $Z_{(GS)i}^{(kj)} = \begin{cases} (Y_i + \alpha S) + \hat{\beta}_{kj}\left(\bar{X} - x_{1i}\right), & p^* = \frac{\beta}{\alpha+\beta} \\ (Y_i - \beta S) + \hat{\beta}_{kj}\left(\bar{X} - x_{2i}\right), & 1 - p^* = \frac{\alpha}{\alpha+\beta} \end{cases}$ where $E(S) = \theta \neq 0$ |
| Azeem et al. (2023) $Z_{(AZ)i} = \begin{cases} Y_i + \alpha S, & p^* = \frac{\beta}{\alpha+\beta} \\ Y_i - \beta S, & 1 - p^* = \frac{\alpha}{\alpha+\beta} \end{cases}$ | $Z_{(AZ)i}^{(kj)} = \begin{cases} (Y_i + \alpha S) + \hat{\beta}_{kj}\left(\bar{X} - x_{1i}\right), & p^* = \frac{\beta}{\alpha+\beta} \\ (Y_i - \beta S) + \hat{\beta}_{kj}\left(\bar{X} - x_{2i}\right), & 1 - p^* = \frac{\alpha}{\alpha+\beta} \end{cases}$ where $E(S) = 0$ |

Since for all $\hat{\mu} \in (\hat{\mu}_G, \hat{\mu}_B, \hat{\mu}_{GS}, \hat{\mu}_{DP}, \hat{\mu}_{ZH}, \hat{\mu}_{AZ})$, $E(\hat{\mu}) = \mu_y$. Then, $E(\hat{\mu}^{(kj)})$ can be defined as in Eq. (41).

$$E\left(\hat{\mu}^{(kj)}\right) = E\left(\frac{1}{n}\sum_{i=1}^{n} Z_i^{(kj)}\right) \tag{41}$$

$$E\left(\hat{\mu}^{(kj)}\right) = \frac{1}{n}\sum_{i=1}^{n}\left(E\left(\theta(y)_1 + \hat{\beta}_{2j}(\mu_x - X_1)\right)p^* + E\left(\theta(y)_2 + \hat{\beta}_{2j}(\mu_x - X_2)(1-p^*)\right)\right) \tag{42}$$

$$E\left(\hat{\mu}^{(kj)}\right) = E\left(\theta(y)_1 p^* + \theta(y)_2 (1-p^*)\right) + E\left(\beta_{kj}(\mu_x - X_1)p^* + \beta_{kj}(\mu_x - X_2)(1-p^*)\right) \tag{43}$$

$$E\left(\hat{\mu}^{(kj)}\right) = \mu_y + \beta_{kj}(\mu_x - \mu_x) = \mu_y \tag{44}$$

Hence, the proof:

**Theorem 2:** Given that $\hat{\beta}_{kj}$, $k,j = 1,2$ is unbiased of $\beta_{kj}$, $k,j = 1,2$, then,

$$\text{var}\left(\hat{\mu}^{(kj)}\right) = \frac{1}{n}\left(\text{var}(Z) + \beta_{kj}^2\sigma_x^2 - 2\beta_{kj}\text{cov}(Z\,X)\right) \tag{45}$$

**Proof:** The variance of $\hat{\mu}^{(kj)}$

$$\text{var}\left(\hat{\mu}^{(kj)}\right) = \text{var}\left(\frac{1}{n}\sum_{i=1}^{n} Z_i^{(kj)}\right) = \frac{1}{n^2}\sum_{i=1}^{n}\left(E\left(Z_i^{(kj)2}\right) - \mu_y^2\right) \tag{46}$$

$$\text{var}\left(\hat{\mu}^{(kj)}\right) = \frac{1}{n}\left(E\left(\theta(y)_1 + \hat{\beta}_{kj}(\mu_x - X_1)\right)^2 p^* + E\left(\theta(y)_2 + \hat{\beta}_{kj}(\mu_x - X_2)\right)^2 (1-p^*) - \mu_y^2\right) \tag{47}$$

$$\text{var}\left(\hat{\mu}^{(kj)}\right) = \frac{1}{n}E\left(\begin{array}{c}\{\theta(y)_1\}^2 p^* + \{\theta(y)_2\}^2 (1-p^*) + \beta_{kj}^2\left\{(\mu_x - X_1)^2 p^* + (\mu_x - X_2)^2 (1-p^*)\right\} \\ -2\beta_{2j}\left\{\theta(y)_1(X_1 - \mu_x)p^* + \theta(y)_2(X_2 - \mu_x)(1-p^*)\right\} - \mu_y^2\end{array}\right) \tag{48}$$

Simplify Eq. (48), Eq. (45) is obtained, Hence the proof.

**Theorem 3:** The privacy level of $Z^{(kj)}$ denoted by $\Delta^{(kj)} = E\left(Z^{(kj)} - Y\right)^2$ is given as in Eq. (49).

$$\Delta^{(kj)} = \text{var}(Z) + \beta_{kj}^2\sigma_x^2 + \sigma_y^2 + 2\beta_{kj}\rho_{yx}\sigma_y\sigma_x - 2\left(\beta_{kj}\text{cov}\left(Z^{(kj)}X\right) + \text{cov}\left(Z^{(kj)}Y\right)\right) \tag{49}$$

**Proof:**

$$\Delta^{(kj)} = E\left(\theta(y)_1 + \hat{\beta}_{kj}(\mu_x - X_1) - Y\right)^2 p^* + E\left(\theta(y)_2 + \hat{\beta}_{kj}(\mu_x - X_2) - Y\right)^2 (1-p^*) \tag{50}$$

$$\Delta^{(kj)} = E\left(\begin{array}{c}\{\theta(y)_1\}^2 p^* + \{\theta(y)_2\}^2 (1-p^*) + \beta_{kj}^2\left\{(\mu_x - X_1)^2 p^* + (\mu_x - X_2)^2 (1-p^*)\right\} \\ Y^2 - 2\beta_{kj}\left\{\theta(y)_1(X_1 - \mu_x)p^* + \theta(y)_2(X_2 - \mu_x)(1-p^*)\right\} - 2 \\ \{\theta(y)_1 Y p^* + \theta(y)_2 Y(1-p^*)\} + 2\beta_{kj}\left\{(X_1 - \mu_x)Y p^* + (X_2 - \mu_x)Y(1-p^*)\right\}\end{array}\right) \tag{51}$$

Take expectation of Eq. (51), Eq. (52) is obtained.

$$\Delta^{(kj)} = var\left(Z^{(kj)}\right) + \mu_y^2 + \beta_{kj}^2\sigma_x^2 + \sigma_y^2 + \mu_y^2 - 2\beta_{kj}\,cov\left(Z^{(kj)}X\right) - 2\left(cov\left(Z^{(kj)}Y\right) + \mu_y^2\right)$$
$$+2\beta_{kj}\rho_{yx}\sigma_y\sigma_x \tag{52}$$

Simplify Eq. (52), Eq. (49) is obtained. Hence, the proof.

By using the results of Eqs. (45) and (49), the combined metric of privacy level and efficiency of $\hat{\mu}^{(kj)}$, denoted by $\delta^{(kj)} = \frac{var\left(\hat{\mu}^{(kj)}\right)}{\Delta^{(kj)}}$ is obtained as in Eq. (53).

$$\delta^{(kj)} = \frac{var\left(Z\right) + \beta_{kj}^2\sigma_x^2 - 2\beta_{kj}\,cov\left(Z\,X\right)}{n\left(var\left(Z\right) + \beta_{kj}^2\sigma_x^2 + \sigma_y^2 + 2\beta_{kj}\rho_{yx}\sigma_y\sigma_x - 2\left(\beta_{kj}\,cov\left(Z^{(kj)}\,X\right) + cov\left(Z^{(kj)}\,Y\right)\right)\right)} \tag{53}$$

**Properties of proposed model $Z_{GS}^{(kj)}$.**

(i) The sample mean of the model $Z_{GS}^{(kj)}$, denoted by $\hat{\mu}_{GS}^{(kj)}$ is obtained as in Eq. (54).

$$\hat{\mu}_G^{(kj)} = \frac{1}{n}\sum_{i=1}^n Z_{Gi}^{(jk)} = \hat{\mu}_G + \hat{\beta}_{kj}\left(\mu_x - \bar{x}\right) \tag{54}$$

(ii) Let $Z_{GS}^{(kj)} = Z^{(kj)}$ in Eq. (49), then $cov\left(Z_{GS}^{(kj)}X\right)$ and $cov\left(Z_{GS}^{(kj)}Y\right)$ are obtained as in Eqs. (55) and (56) respectively.

$$cov\left(Z_{GS}^{(kj)}X\right) = E\left\{(Y+\alpha S)(X_1-\mu_x)\frac{\beta}{\alpha+\beta} + (Y-\beta S)(X_2-\mu_x)\frac{\alpha}{\alpha+\beta}\right\} = \rho_{yx}\sigma_y\sigma_x \tag{55}$$

$$cov\left(Z_{GS}^{(kj)}Y\right) = E\left\{(Y+\alpha S)Y\frac{\beta}{\alpha+\beta} + (Y-\beta S)Y\frac{\alpha}{\alpha+\beta}\right\} - \mu_y^2 = \sigma_y^2 \tag{56}$$

(iii) Using the result of Eq. (55), the variance of $\hat{\mu}_{GS}^{(kj)}$ denoted by $var\left(\hat{\mu}_{GS}^{(kj)}\right)$ is obtained as in Eq. (57).

$$var\left(\hat{\mu}_{GS}^{(kj)}\right) = \frac{1}{n}\left(\sigma_y^2 + \alpha\beta\left(\sigma_s^2 + \mu_s^2\right) + \beta_{kj}^2\sigma_x^2 - 2\beta_{kj}\rho_{yx}\sigma_y\sigma_x\right) \tag{57}$$

(iv) Using the results of Eqs. (55)–(57), the privacy level of $\hat{\mu}_{GS}^{(kj)}$ denoted by $\Delta_{GS}^{(kj)}$ is obtained as in Eq. (58).

$$\Delta_{GS}^{(kj)} = \alpha\beta\left(\sigma_s^2 + \mu_s^2\right) + 2\left(\beta_{kj}^2\sigma_x^2 - \beta_{kj}\rho_{yx}\sigma_y\sigma_x\right) \tag{58}$$

(iv) Using the results of Eqs. (57) and (58), the combined metric of privacy level and efficiency of $\hat{\mu}_{GS}^{(kj)}$ denoted by $\delta_{GS}^{(kj)} = \frac{var\left(\hat{\mu}_{GS}^{(kj)}\right)}{\Delta_{GS}^{(kj)}}$ is obtained as in Eq. (59).

$$\delta_{GS}^{(kj)} = \frac{\sigma_y^2 + \alpha\beta\left(\sigma_s^2 + \mu_s^2\right) + \beta_{kj}^2\sigma_x^2 - 2\beta_{kj}\rho_{yx}\sigma_y\sigma_x}{n\left(\alpha\beta\left(\sigma_s^2 + \mu_s^2\right) + 2\left(\beta_{kj}^2\sigma_x^2 - \beta_{kj}\rho_{yx}\sigma_y\sigma_x\right)\right)} \tag{59}$$

Members of the Proposed Calibrated Optional Quantitative RRT Model $Z_{GS}^{(kj)}$ are presented in Eqs. (60)–(63).

$$Z_{(GS)i}^{(11)} = \begin{cases} Y_i + \alpha S + \dfrac{\sum\limits_{i \in G_1} x_i(Y_i + \alpha S) + \sum\limits_{i \in G_2} x_i(Y_i - \beta S)}{\sum\limits_{i \in G_1} x_{1i}^2 + \sum\limits_{i \in G_2} x_{2i}^2}\left(\bar{X} - x_{1i}\right), & p^* = \dfrac{\alpha}{\alpha + \beta} \\[4mm] Y_i - \beta S + \dfrac{\sum\limits_{i \in G_1} x_i(Y_i + \alpha S) + \sum\limits_{i \in G_2} x_i(Y_i - \beta S)}{\sum\limits_{i \in G_1} x_{1i}^2 + \sum\limits_{i \in G_2} x_{2i}^2}\left(\bar{X} - x_{2i}\right), & 1 - p^* = \dfrac{\beta}{\alpha + \beta} \end{cases}$$

(60)

$$Z_{(GS)i}^{(12)} = \begin{cases} Y_i + \alpha S + \dfrac{\sum\limits_{i \in G_1} (Y_i + \alpha S) + \sum\limits_{i \in G_2} (Y_i - \beta S)}{\sum\limits_{i \in G_1} x_{1i} + \sum\limits_{i \in G_2} x_{2i}}\left(\bar{X} - x_{1i}\right), & p^* = \dfrac{\alpha}{\alpha + \beta} \\[4mm] Y_i - \beta S + \dfrac{\sum\limits_{i \in G_1} (Y_i + \alpha S) + \sum\limits_{i \in G_2} (Y_i - \beta S)}{\sum\limits_{i \in G_1} x_{1i} + \sum\limits_{i \in G_2} x_{2i}}\left(\bar{X} - x_{2i}\right), & 1 - p^* = \dfrac{\beta}{\alpha + \beta} \end{cases}$$

(61)

$$Z_{(GS)i}^{(21)} = \begin{cases} Y_i + \alpha S + \dfrac{\sum\limits_{i \in G_1} (x_i - \bar{x})(Y_i + \alpha S) + \sum\limits_{i \in G_2} (x_i - \bar{x})(Y_i - \beta S)}{\sum\limits_{i \in G_1} x_i^2 + \sum\limits_{i \in G_2} x_i^2 - n\bar{x}^2}\left(\bar{X} - x_i\right), & p^* = \dfrac{\alpha}{\alpha + \beta} \\[4mm] Y_i - \beta S + \dfrac{\sum\limits_{i \in G_1} (x_i - \bar{x})(Y_i + \alpha S) + \sum\limits_{i \in G_2} (x_i - \bar{x})(Y_i - \beta S)}{\sum\limits_{i \in G_1} x_i^2 + \sum\limits_{i \in G_2} x_i^2 - n\bar{x}^2}\left(\bar{X} - x_{2i}\right), & 1 - p^* = \dfrac{\beta}{\alpha + \beta} \end{cases}$$

(62)

$$Z_{(GS)i}^{(22)} = \begin{cases} Y_i + \alpha S + \dfrac{\hat{\mu}_{GS}\left(\sum\limits_{i \in G_1} \left(\frac{1}{x_i} - \frac{x_i}{Y_i + \alpha S}\right) + \sum\limits_{i \in G_2} \left(\frac{1}{x_i} - \frac{x_i}{Y_i - \beta S}\right)\right)}{\bar{x}\left(\sum\limits_{i \in G_1} \frac{1}{x_i} + \sum\limits_{i \in G_2} \frac{1}{x_i}\right) - 1}\left(\bar{X} - x_{1i}\right), & p^* = \dfrac{\alpha}{\alpha + \beta} \\[6mm] Y_i - \beta S + \dfrac{\hat{\mu}_{GS}\left(\sum\limits_{i \in G_1} \left(\frac{1}{x_i} - \frac{x_i}{Y_i + \alpha S}\right) + \sum\limits_{i \in G_2} \left(\frac{1}{x_i} - \frac{x_i}{Y_i - \beta S}\right)\right)}{\bar{x}\left(\sum\limits_{i \in G_1} \frac{1}{x_i} + \sum\limits_{i \in G_2} \frac{1}{x_i}\right) - 1}\left(\bar{X} - x_{2i}\right), & 1 - p^* = \dfrac{\beta}{\alpha + \beta} \end{cases}$$

(63)

**Properties of proposed model $Z_{AZ}^{(kj)}$.**

(i) The sample mean of the model $Z_{AZ}^{(kj)}$, denoted by $\hat{\mu}_{AZ}^{(kj)}$ is obtained as in Eq. (64).

$$\hat{\mu}_{AZ}^{(kj)} = \frac{1}{n}\sum_{i=1}^{n} Z_{AZi}^{(jk)} = \hat{\mu}_{AZ} + \hat{\beta}_{kj}\left(\mu_x - \bar{x}\right)$$

(64)

(ii) Let $Z_{AZ}^{(kj)} = Z^{(kj)}$ in Eq. (49), then $\text{cov}\left(Z_{AZ}^{(kj)}X\right)$ and $\text{cov}\left(Z_{AZ}^{(kj)}Y\right)$ are obtained as in Eqs. (65) and (66) respectively.

$$\text{cov}\left(Z_{AZ}^{(kj)}X\right) = E\left\{(Y + \alpha S)(X_1 - \mu_x)\frac{\beta}{\alpha + \beta} + (Y - \beta S)(X_2 - \mu_x)\frac{\alpha}{\alpha + \beta}\right\} = \rho_{yx}\sigma_y\sigma_x$$

(65)

$$\text{cov}\left(Z_{AZ}^{(kj)}Y\right) = E\left\{(Y + \alpha S)Y\frac{\beta}{\alpha + \beta} + (Y - \beta S)Y\frac{\alpha}{\alpha + \beta}\right\} - \mu_y^2 = \sigma_y^2$$

(66)

(iii) Using the result of Eq. (65), the variance of $\hat{\mu}_{AZ}^{(kj)}$ denoted by $\text{var}\left(\hat{\mu}_{AZ}^{(kj)}\right)$ is obtained as in Eq. (67).

$$\text{var}\left(\hat{\mu}_{AZ}^{(kj)}\right) = \frac{1}{n}\left(\sigma_y^2 + \alpha\beta\sigma_s^2 + \beta_{kj}^2\sigma_x^2 - 2\beta_{kj}\rho_{yx}\sigma_y\sigma_x\right)$$

(67)

(iv) Using the results of Eqs. (65) and (66), the privacy level of $\hat{\mu}_{AZ}^{(kj)}$ denoted by $\Delta_{AZ}^{(kj)}$ is obtained as in Eq. (68)

$$\Delta_{AZ}^{(kj)} = \alpha\beta\sigma_s^2 + 2\left(\beta_{kj}^2\sigma_x^2 - \beta_{kj}\rho_{yx}\sigma_y\sigma_x\right) \tag{68}$$

(v) Using the results of Eqs. (67) and (68), the combined metric of privacy level and efficiency of $\hat{\mu}_{GS}^{(kj)}$ denoted by $\delta_{AZ}^{(kj)} = \frac{\text{var}\left(\hat{\mu}_{AZ}^{(kj)}\right)}{\Delta_{AZ}^{(kj)}}$ is obtained as in Eq. (69).

$$\delta_{AZ}^{(kj)} = \frac{\sigma_y^2 + \alpha\beta\sigma_s^2 + \beta_{kj}^2\sigma_x^2 - 2\beta_{kj}\rho_{yx}\sigma_y\sigma_x}{n\left(\alpha\beta\sigma_s^2 + 2\left(\beta_{kj}^2\sigma_x^2 - \beta_{kj}\rho_{yx}\sigma_y\sigma_x\right)\right)} \tag{69}$$

Members of the Proposed Calibrated Optional Quantitative RRT Model $Z_{AZ}^{(kj)}$ are presented in Eqs. (70)–(73).

$$Z_{(AZ)i}^{(11)} = \begin{cases} Y_i + \alpha S + \dfrac{\sum\limits_{i\in G_1} x_i(Y_i+\alpha S) + \sum\limits_{i\in G_2} x_i(Y_i-\beta S)}{\sum\limits_{i\in G_1} x_{1i}^2 + \sum\limits_{i\in G_2} x_{2i}^2}\left(\bar{X}-x_{1i}\right), & p^* = \dfrac{\alpha}{\alpha+\beta} \\[4mm] Y_i - \beta S + \dfrac{\sum\limits_{i\in G_1} x_i(Y_i+\alpha S) + \sum\limits_{i\in G_2} x_i(Y_i-\beta S)}{\sum\limits_{i\in G_1} x_{1i}^2 + \sum\limits_{i\in G_2} x_{2i}^2}\left(\bar{X}-x_{2i}\right), & 1-p^* = \dfrac{\beta}{\alpha+\beta} \end{cases} \tag{70}$$

$$Z_{(AZ)i}^{(12)} = \begin{cases} Y_i + \alpha S + \dfrac{\sum\limits_{i\in G_1} (Y_i+\alpha S) + \sum\limits_{i\in G_2} (Y_i-\beta S)}{\sum\limits_{i\in G_1} x_{1i} + \sum\limits_{i\in G_2} x_{2i}}\left(\bar{X}-x_{1i}\right), & p^* = \dfrac{\alpha}{\alpha+\beta} \\[4mm] Y_i - \beta S + \dfrac{\sum\limits_{i\in G_1} (Y_i+\alpha S) + \sum\limits_{i\in G_2} (Y_i-\beta S)}{\sum\limits_{i\in G_1} x_{1i} + \sum\limits_{i\in G_2} x_{2i}}\left(\bar{X}-x_{2i}\right), & 1-p^* = \dfrac{\beta}{\alpha+\beta} \end{cases} \tag{71}$$

$$Z_{(AZ)i}^{(21)} = \begin{cases} Y_i + \alpha S + \dfrac{\sum\limits_{i\in G_1} (x_i-\bar{x})(Y_i+\alpha S) + \sum\limits_{i\in G_2} (x_i-\bar{x})(Y_i-\beta S)}{\sum\limits_{i\in G_1} x_i^2 + \sum\limits_{i\in G_2} x_i^2 - n\bar{x}^2}\left(\bar{X}-x_i\right), & p^* = \dfrac{\alpha}{\alpha+\beta} \\[4mm] Y_i - \beta S + \dfrac{\sum\limits_{i\in G_1} (x_i-\bar{x})(Y_i+\alpha S) + \sum\limits_{i\in G_2} (x_i-\bar{x})(Y_i-\beta S)}{\sum\limits_{i\in G_1} x_i^2 + \sum\limits_{i\in G_2} x_i^2 - n\bar{x}^2}\left(\bar{X}-x_{2i}\right), & 1-p^* = \dfrac{\beta}{\alpha+\beta} \end{cases} \tag{72}$$

$$Z_{(AZ)i}^{(22)} = \begin{cases} Y_i + \alpha S + \dfrac{\hat{\mu}_{AZ}\left(\sum\limits_{i\in G_1} \left(\frac{1}{x_i}-\frac{x_i}{Y_i+\alpha S}\right) + \sum\limits_{i\in G_2} \left(\frac{1}{x_i}-\frac{x_i}{Y_i-\beta S}\right)\right)}{\bar{x}\left(\sum\limits_{i\in G_1} \frac{1}{x_i} + \sum\limits_{i\in G_2} \frac{1}{x_i}\right)-1}\left(\bar{X}-x_{1i}\right), & p^* = \dfrac{\alpha}{\alpha+\beta} \\[4mm] Y_i - \beta S + \dfrac{\hat{\mu}_{AZ}\left(\sum\limits_{i\in G_1} \left(\frac{1}{x_i}-\frac{x_i}{Y_i+\alpha S}\right) + \sum\limits_{i\in G_2} \left(\frac{1}{x_i}-\frac{x_i}{Y_i-\beta S}\right)\right)}{\bar{x}\left(\sum\limits_{i\in G_1} \frac{1}{x_i} + \sum\limits_{i\in G_2} \frac{1}{x_i}\right)-1}\left(\bar{X}-x_{2i}\right), & 1-p^* = \dfrac{\beta}{\alpha+\beta} \end{cases} \tag{73}$$

**Theoretical efficiency comparison**

This subsection presents theoretical comparisons of the proposed calibrated optional RRT models with their existing counterparts.

i. The Proposed C-RRT Model $Z_{GS}^{(kj)}$ is more efficient than RRT Model $Z_{GS}$ if $\text{var}\left(\hat{\mu}_{GS}\right)/\text{var}\left(\hat{\mu}_{GS}^{(kj)}\right) > 1$. Therefore, the condition in Eq. (74) is obtained.

$$\beta_{kj} < \frac{2\rho_{yx}\sigma_y}{\sigma_x} \tag{74}$$

ii. The Proposed C-RRT Model $Z_{AZ}^{(kj)}$ is more efficient than RRT Model $Z_{AZ}$ if $\text{var}\left(\hat{\mu}_{AZ}\right)/\text{var}\left(\hat{\mu}_{AZ}^{(kj)}\right) > 1$. Therefore, the condition in Eq. (75) is obtained.

$$\beta_{kj} < \frac{2\rho_{yx}\sigma_y}{\sigma_x} \tag{75}$$

## Results and discussion

### Simulation study

In this subsection, simulation studies were conducted using some probability distributions defined in Table 2 to assess the performance of the proposed models with respect to existing models under study. Data of size 1000 units were generated for the study population using function defined in Table 2. Sample of size 100 units were selected using simple random sampling without replacement to compute the biases, efficiency, percentage relative efficiency, privacy level and combined metric of efficiency and privacy of the estimators using Eqs. (76)–(80) respectively. The processes of sampling and computation procedures were conducted 100 times and averages of the results were computed as presented in Tables 3–8.

$$Bias\,(Z) = E\,(Z - \mu_Z) \tag{76}$$

$$Var\,(Z) = E(Z - \mu_Z)^2 \tag{77}$$

$$PRE\,(Z) = \frac{Var\,(Y)}{Var\,(Z)} \times 100 \tag{78}$$

$$\Delta_Z = E(Z - Y)^2 \tag{79}$$

$$\delta_Z = \frac{Var\,(Z)}{\Delta_Z} \tag{80}$$

i. Tables 3, 5 and 7 present the numerical results of the estimate population mean for the existing and proposed estimators as well as the variance, PRE, privacy level and Combined Metric of Efficiency and Privacy level of the estimators for population 1, 2, and 3 when $\alpha = 1$, $\beta = 4$ and $\alpha = 2$, $\beta = 3$ through simulation studies. Similarly, Tables 4, 6 and 8 present the numerical results of the estimate population mean for the existing and proposed estimators as well as the variance, PRE, privacy level and Combined Metric of Efficiency and Privacy level of the estimators for population 1, 2, and 3 when $\alpha = 3$, $\beta = 2$ and $\alpha = 4$, $\beta = 1$ through simulation studies

**Table 2. Distributions of non-linear populations used for empirical study.**

| Population | Auxiliary variable $x$ | Study variable $y$ |
|---|---|---|
| I | $X \sim \exp\,(1)$ | $Y_i = X_i + \varepsilon_i, \quad \varepsilon \sim N\,(0,1)$ |
| II | $X \sim \log normal\,(5,8)$ | |
| III | $X \sim weibull\,(1,7)$ | |

**Table 3. Bias, variance, PRE, privacy level and combined metric of efficiency and privacy level when $\alpha = 1$, $\beta = 4$ and $\alpha = 2$, $\beta = 3$ using population I.**

| Models $Z_G$ | Estimate $\hat{\mu}_y$ | Var $(Z)$ | PRE $(Z)$ | Privacy Level $\Delta_z$ | Combined Metric $\delta_z = $ Var $(Z) / \Delta_z$ |
|---|---|---|---|---|---|
| $\alpha = 1$, $\beta = 4$ | | | | | |
| The value of Parameter being estimated $\bar{Y} = \mu_Y = 10.19$ | | | | | |
| Y | 11.4454 | 132.5471 | 100 | 0 | NA |
| $Z_{GS}$ | 8.439245 | 142.4927 | 93.02024 | 205.1096 | 0.694715 |
| $Z_{AZ}$ | 11.20666 | 179.8739 | 73.68887 | 223.889 | 0.8034067 |
| $Z_{GS}^{(11)}$ | 10.33986 | 42.3546 | 312.9461 | 125.9513 | 0.3362777 |
| $Z_{GS}^{(12)}$ | 10.51614 | 50.11667 | 264.477 | 131.0538 | 0.3824129 |
| $Z_{GS}^{(21)}$ | 10.17011 | 39.35342 | 336.8121 | 125.5699 | 0.3133986 |
| $Z_{GS}^{(22)}$ | 10.90476 | 83.94719 | 157.8934 | 159.2409 | 0.5271712 |
| $Z_{AZ}^{(11)}$ | 10.71879 | 46.19781 | 286.9121 | 174.7877 | 0.2643081 |
| $Z_{AZ}^{(12)}$ | 10.9883 | 55.94104 | 236.9407 | 162.955 | 0.3432913 |
| $Z_{AZ}^{(21)}$ | 10.60892 | 42.82045 | 309.5415 | 171.4861 | 0.2497022 |
| $Z_{AZ}^{(22)}$ | 11.68845 | 137.7176 | 96.24553 | 205.5289 | 0.6700646 |
| $\alpha = 1$, $\beta = 4$ | | | | | |
| Y | 11.4454 | 132.5471 | 100 | 0 | NA |
| $Z_{GS}$ | 10.4024 | 142.9459 | 92.72534 | 134.2127 | 1.06507 |
| $Z_{AZ}$ | 12.14269 | 161.1113 | 82.27049 | 183.3065 | 0.8789176 |
| $Z_{GS}^{(11)}$ | 10.19277 | 33.64866 | 393.9149 | 133.0647 | 0.2528744 |
| $Z_{GS}^{(12)}$ | 10.25116 | 34.65219 | 382.507 | 130.1785 | 0.2661899 |
| $Z_{GS}^{(21)}$ | 10.13655 | 33.17302 | 399.5629 | 136.3413 | 0.2433087 |
| $Z_{GS}^{(22)}$ | 10.56587 | 49.00361 | 270.4843 | 123.6807 | 0.3962106 |
| $Z_{AZ}^{(11)}$ | 10.17887 | 21.91696 | 604.7695 | 154.5233 | 0.141836 |
| $Z_{AZ}^{(12)}$ | 10.29915 | 22.40059 | 591.7124 | 145.1221 | 0.1543568 |
| $Z_{AZ}^{(21)}$ | 10.20878 | 21.7915 | 608.2514 | 151.3275 | 0.1440022 |
| $Z_{AZ}^{(22)}$ | 12.97847 | 605.6113 | 21.8865 | 533.737 | 1.134662 |

ii. Tables 3–8 present the results of the estimate of the population parameter for the proposed and existing estimators and the results revealed that the estimators of proposed models $Z_{GS}^{(11)}$, $Z_{GS}^{(12)}$, $Z_{GS}^{(21)}$, $Z_{GS}^{(22)}$, $Z_{AZ}^{(11)}$, $Z_{AZ}^{(12)}$, $Z_{AZ}^{(21)}$ and $Z_{AZ}^{(22)}$ produced estimates that are closer to the corresponding population parameter than the existing estimators except in few cases. For instance, under Population I, when estimating a population value of 11.4454, the C-TORRT estimators gave values like 10.20878, compared to 12.14269 from Azeem et al (2023). This indicates better accuracy and less systematic error in the proposed models. This implies that the proposed models are more robust than the existing models.

iii. The results of variances of the estimators of the proposed and existing models are presented in Tables 3–8 and the results revealed that the estimators of proposed models $Z_{GS}^{(11)}$, $Z_{GS}^{(12)}$, $Z_{GS}^{(21)}$, $Z_{GS}^{(22)}$, $Z_{AZ}^{(11)}$, $Z_{AZ}^{(12)}$, $Z_{AZ}^{(21)}$ and $Z_{AZ}^{(22)}$ have

**Table 4. Bias, variance, PRE, privacy level and combined metric of efficiency and privacy level when $\alpha = 3$, $\beta = 2$ and $\alpha = 4$, $\beta = 1$ using population I.**

| Models $Z_G$ | Estimate $\hat{\mu}_y$ | Var $(Z)$ | PRE $(Z)$ | Privacy Level $\Delta_Z$ | Combined Metric $\delta_Z = Var(Z)/\Delta_Z$ |
|---|---|---|---|---|---|
| $\alpha = 3$, $\beta = 2$ | | | | | |
| The value of Parameter being estimated $\bar{Y} = \mu_Y = 10.19$ | | | | | |
| Y | 11.4454 | 132.5471 | 100 | 0 | NA |
| $Z_{GS}$ | 12.32314 | 149.5401 | 88.63646 | 157.0149 | 0.9523948 |
| $Z_{AZ}$ | 11.61547 | 150.0307 | 88.34664 | 136.4856 | 1.099242 |
| $Z_{GS}^{(11)}$ | 10.43409 | 21.76704 | 608.9347 | 222.4369 | 0.09785718 |
| $Z_{GS}^{(12)}$ | 10.39366 | 21.85759 | 606.4122 | 228.0839 | 0.09583137 |
| $Z_{GS}^{(21)}$ | 10.47302 | 21.91514 | 604.8197 | 217.2373 | 0.1008811 |
| $Z_{GS}^{(22)}$ | 11.67459 | 140.0028 | 94.67456 | 171.765 | 0.8150836 |
| $Z_{AZ}^{(11)}$ | 9.491216 | 25.40491 | 521.7381 | 162.704 | 0.1561419 |
| $Z_{AZ}^{(12)}$ | 9.642512 | 27.63385 | 479.6548 | 181.1327 | 0.1525613 |
| $Z_{AZ}^{(21)}$ | 9.852679 | 24.62443 | 538.2747 | 155.4096 | 0.1584486 |
| $Z_{AZ}^{(22)}$ | 11.09218 | 143.7202 | 92.22576 | 142.3444 | 1.009665 |
| $\alpha = 4$, $\beta = 1$ | | | | | |
| Y | 11.4454 | 132.5471 | 100 | 0 | NA |
| $Z_{GS}$ | 14.35088 | 141.6709 | 93.55986 | 49.57972 | 2.857436 |
| $Z_{AZ}$ | 12.44746 | 162.8208 | 81.40673 | 80.76804 | 2.015906 |
| $Z_{GS}^{(11)}$ | 10.40368 | 19.53398 | 678.5464 | 163.7384 | 0.1192999 |
| $Z_{GS}^{(12)}$ | 10.22296 | 24.01465 | 551.9427 | 199.7399 | 0.1202296 |
| $Z_{GS}^{(21)}$ | 10.57771 | 19.92078 | 665.371 | 133.833 | 0.148848 |
| $Z_{GS}^{(22)}$ | 11.92899 | 179.8941 | 73.68063 | 60.65544 | 2.965836 |
| $Z_{AZ}^{(11)}$ | 9.87964 | 7.072311 | 1874.169 | 177.3626 | 0.03987487 |
| $Z_{AZ}^{(12)}$ | 9.63038 | 16.59664 | 798.6378 | 244.7269 | 0.067817 |
| $Z_{AZ}^{(21)}$ | 9.997561 | 4.45392 | 2975.964 | 163.2418 | 0.02728418 |
| $Z_{AZ}^{(22)}$ | 11.1286 | 96.10285 | 137.9221 | 42.98718 | 2.235616 |

minimum variances compared to that of the existing estimators except in few cases. For instance, under Population I, the proposed estimator had variance 21.76704, compared to 142.4927 for Azeem et al. (2023). Under Population II, a variance of 4.3098 was observed for a C-TORRT estimator, compared to 21.5316 for Gjestvang & Singh. These reductions indicate substantial gains in statistical efficiency. This implies that the proposed models are more efficient than the existing models.

iv. Similarly, Tables 3–8 present the results of the percentage relative efficiency (PRE) for the proposed and existing estimators and the results revealed that the estimators of proposed models $Z_{GS}^{(11)}$, $Z_{GS}^{(12)}$, $Z_{GS}^{(21)}$, $Z_{GS}^{(22)}$, $Z_{AZ}^{(11)}$, $Z_{AZ}^{(12)}$, $Z_{AZ}^{(21)}$ and $Z_{AZ}^{(22)}$ have higher PRE values compared to the existing estimators with exception of few cases. Under Population I,

**Table 5. Bias, variance, PRE, privacy level and combined metric of efficiency and privacy level when $\alpha = 1$, $\beta = 4$ and $\alpha = 2$, $\beta = 3$ using population ii.**

| Models $Z_G$ | Estimate $\hat{\mu}_Y$ | Var $(Z)$ | PRE $(Z)$ | Privacy Level $\Delta_Z$ | Combined Metric $\delta_Z = Var(Z)/\Delta_Z$ |
|---|---|---|---|---|---|
| $\alpha = 1$, $\beta = 4$ | | | | | |
| The value of Parameter being estimated $\bar{Y} = \mu_Y = 684572361$ | | | | | |
| Y | 61157008 | 1.520689e+17 | 100 | 0 | NA |
| $Z_{GS}$ | 61157006 | 1.520689e+17 | 100 | 1.897585e+17 | 0.801381 |
| $Z_{AZ}$ | 61157009 | 1.520689e+17 | 100 | 1.51675e+17 | 1.002597 |
| $Z_{GS}^{(11)}$ | 701104154 | 1.780679e+16 | 853.9939 | 5.057788e+17 | 0.03520667 |
| $Z_{GS}^{(12)}$ | 701104129 | 1.780679e+16 | 853.9937 | 5.057787e+17 | 0.03520668 |
| $Z_{GS}^{(21)}$ | 701104155 | 1.780678e+16 | 853.9939 | 5.057788e+17 | 0.03520667 |
| $Z_{GS}^{(22)}$ | 701080279 | 1.781028e+16 | 853.8262 | 5.057502e+17 | 0.03521557 |
| $Z_{AZ}^{(11)}$ | 687652225 | 7.263152e+13 | 209370.3 | 6.178557e+17 | 0.0001175542 |
| $Z_{AZ}^{(12)}$ | 687652196 | 7.263152e+13 | 209370.3 | 6.178556e+17 | 0.0001175542 |
| $Z_{AZ}^{(21)}$ | 687652226 | 7.263152e+13 | 209370.3 | 6.178557e+17 | 0.0001175542 |
| $Z_{AZ}^{(22)}$ | 687629576 | 7.263229e+13 | 209368.1 | 6.178192e+17 | 0.0001175624 |
| $\alpha = 2$, $\beta = 3$ | | | | | |
| Y | 61157008 | 1.520689e+17 | 100 | 0 | NA |
| $Z_{GS}$ | 61157008 | 1.520689e+17 | 100 | 1.558083e+17 | 0.9759998 |
| $Z_{AZ}$ | 61157008 | 1.520689e+17 | 100 | 1.556728e+17 | 0.976849 |
| $Z_{GS}^{(11)}$ | 701086170 | 1.775991e+16 | 856.248 | 4.985093e+17 | 0.03562603 |
| $Z_{GS}^{(12)}$ | 701086164 | 1.775991e+16 | 856.2479 | 4.985093e+17 | 0.03562603 |
| $Z_{GS}^{(21)}$ | 701086170 | 1.775991e+16 | 856.248 | 4.985093e+17 | 0.03562603 |
| $Z_{GS}^{(22)}$ | 701055938 | 1.776433e+16 | 856.0347 | 4.984719e+17 | 0.03563758 |
| $Z_{AZ}^{(11)}$ | 687365834 | 4.451724e+13 | 341595.4 | 6.103572e+17 | 7.293638e-05 |
| $Z_{AZ}^{(12)}$ | 687365827 | 4.451724e+13 | 341595.4 | 6.103571e+17 | 7.293638e-05 |
| $Z_{AZ}^{(21)}$ | 687365834 | 4.451724e+13 | 341595.4 | 6.103572e+17 | 7.293638e-05 |
| $Z_{AZ}^{(22)}$ | 687336411 | 4.45163e+13 | 341602.7 | 6.103102e+17 | 7.294044e-05 |

C-TORRT models showed PRE > 600, while Azeem's model had PRE = 93.02. Under Population II, C-TORRT achieved PRE = 499.6, again vastly outperforming existing techniques. These values confirm that C-TORRT models require fewer samples to achieve the same precision. These results imply that with exception of few cases, the proposed models are more precise (efficiency gains) in estimating sensitive information compared to the existing models.

v. Tables 3–8 present the results of the privacy level of the estimators of the proposed and existing models and the results revealed that the estimators of proposed models $Z_{GS}^{(11)}$, $Z_{GS}^{(12)}$, $Z_{GS}^{(21)}$, $Z_{GS}^{(22)}$, $Z_{AZ}^{(11)}$, $Z_{AZ}^{(12)}$, $Z_{AZ}^{(21)}$ and $Z_{AZ}^{(22)}$ have higher privacy level values than the existing estimators except in few cases. For instance, in Population I, a C-TORRT model had privacy level = 228.0839, while Azeem's model had 205.1096. In real-life data, C-TORRT had privacy levels around 817.28, which was higher than corresponding models (813.4977). This reflects improved confidentiality

**Table 6. Bias, variance, pre, privacy level and combined metric of efficiency and privacy level when $\alpha = 3$, $\beta = 2$ and $\alpha = 4$, $\beta = 1$ using population II.**

| Models $Z_G$ | Estimate $\hat{\mu}_Y$ | Var $(Z)$ | PRE $(Z)$ | Privacy Level $\Delta_Z$ | Combined Metric $\delta_Z = Var(Z)/\Delta_Z$ |
|---|---|---|---|---|---|
| $\alpha = 3$, $\beta = 2$ | | | | | |
| The value of Parameter being estimated $\bar{Y} = \mu_Y = 684572361$ | | | | | |
| Y | 61157008 | 1.520689e+17 | 100 | 0 | NA |
| $Z_{GS}$ | 61157009 | 1.520689e+17 | 100 | 1.554861e+17 | 0.9780223 |
| $Z_{AZ}$ | 61157009 | 1.520689e+17 | 100 | 3.433575e+16 | 4.428879 |
| $Z_{GS}^{(11)}$ | 701055572 | 1.775954e+16 | 856.2657 | 5.132514e+17 | 0.03460203 |
| $Z_{GS}^{(12)}$ | 701055583 | 1.775954e+16 | 856.2658 | 5.132514e+17 | 0.03460203 |
| $Z_{GS}^{(21)}$ | 701055572 | 1.775954e+16 | 856.2657 | 5.132514e+17 | 0.03460203 |
| $Z_{GS}^{(22)}$ | 701033125 | 1.776283e+16 | 856.1069 | 5.13223e+17 | 0.03461036 |
| $Z_{AZ}^{(11)}$ | 687747391 | 2.093727e+13 | 726306.9 | 5.345495e+17 | 3.916807e-05 |
| $Z_{AZ}^{(12)}$ | 687747404 | 2.093727e+13 | 726306.9 | 5.345495e+17 | 3.916806e-05 |
| $Z_{AZ}^{(21)}$ | 687747390 | 2.093727e+13 | 726307 | 5.345495e+17 | 3.916806e-05 |
| $Z_{AZ}^{(22)}$ | 687724414 | 2.093655e+13 | 726332.1 | 5.345112e+17 | 3.916952e-05 |
| $\alpha = 4$, $\beta = 1$ | | | | | |
| Y | 61157008 | 1.520689e+17 | 100 | 0 | NA |
| $Z_{GS}$ | 61157011 | 1.520689e+17 | 100 | 1.225898e+17 | 1.240469 |
| $Z_{AZ}$ | 61157008 | 1.520689e+17 | 100 | 1.099497e+15 | 138.3076 |
| $Z_{GS}^{(11)}$ | 701056600 | 1.775951e+16 | 856.2671 | 5.079583e+17 | 0.03496254 |
| $Z_{GS}^{(12)}$ | 701056631 | 1.775951e+16 | 856.2673 | 5.079584e+17 | 0.03496252 |
| $Z_{GS}^{(21)}$ | 701056599 | 1.775951e+16 | 856.2671 | 5.079583e+17 | 0.03496254 |
| $Z_{GS}^{(22)}$ | 701040912 | 1.776181e+16 | 856.156 | 5.079378e+17 | 0.03496848 |
| $Z_{AZ}^{(11)}$ | 687661642 | 2.354562e+13 | 645847.6 | 5.455247e+17 | 4.316143e-05 |
| $Z_{AZ}^{(12)}$ | 687661672 | 2.354561e+13 | 645847.9 | 5.455247e+17 | 4.31614e-05 |
| $Z_{AZ}^{(21)}$ | 687661641 | 2.354562e+13 | 645847.6 | 5.455247e+17 | 4.316143e-05 |
| $Z_{AZ}^{(22)}$ | 687643427 | 2.355244e+13 | 645660.7 | 5.45493e+17 | 4.317643e-05 |

safeguards, vital in sensitive surveys. This implies that the privacy of the respondents are more protected in relation to sensitive information under proposed models than under the existing models.

vi. The results of combined metric of efficiency and privacy level of the estimators of the proposed models $Z_{GS}^{(11)}$, $Z_{GS}^{(12)}$, $Z_{GS}^{(21)}$, $Z_{GS}^{(22)}$, $Z_{AZ}^{(11)}$, $Z_{AZ}^{(12)}$, $Z_{AZ}^{(21)}$, $Z_{AZ}^{(22)}$ and that of the existing models are presented in Tables 3–8 and the results revealed that the estimators of proposed models have minimum values of combine metric for efficiency and privacy level compared to that of the existing models except in few cases. For instance, the proposed C-TORRT models had 0.09 or lower, while existing models ranged up to 0.80 or higher. This implies that the proposed models have better balancing of efficiency and privacy level compared to the existing models.

**Table 7. Bias, variance, PRE, privacy level and combined metric of efficiency and privacy level when $\alpha = 1$, $\beta = 4$ and $\alpha = 2$, $\beta = 3$ using population II.**

| Models $Z_G$ | Estimate $\hat{\mu}_y$ | Var $(Z)$ | PRE $(Z)$ | Privacy Level $\Delta_Z$ | Combined Metric $\delta_Z = Var(Z)/\Delta_Z$ |
|---|---|---|---|---|---|
| $\alpha = 1$, $\beta = 4$ | | | | | |
| The value of Parameter being estimated $\bar{Y} = \mu_Y = 6.97$ | | | | | |
| Y | 5.630163 | 21.53163 | 100 | 0 | NA |
| $Z_{GS}$ | 2.912156 | 27.02452 | 79.67443 | 33.94066 | 0.7962284 |
| $Z_{AZ}$ | 5.524937 | 33.14033 | 64.97109 | 51.00787 | 0.64971 |
| $Z_{GS}^{(11)}$ | 6.099989 | 8.921042 | 241.3578 | 26.72859 | 0.333764 |
| $Z_{GS}^{(12)}$ | 5.866188 | 10.26508 | 209.7561 | 25.82362 | 0.3975075 |
| $Z_{GS}^{(21)}$ | 6.444077 | 9.440713 | 228.0721 | 30.75702 | 0.3069449 |
| $Z_{GS}^{(22)}$ | 5.428817 | 16.46671 | 130.7585 | 28.11156 | 0.5857629 |
| $Z_{AZ}^{(11)}$ | 6.374165 | 7.502558 | 286.9905 | 32.04715 | 0.23411 |
| $Z_{AZ}^{(12)}$ | 6.160632 | 10.65443 | 202.0909 | 29.29417 | 0.3637048 |
| $Z_{AZ}^{(21)}$ | 6.892808 | 7.227995 | 297.8922 | 35.24201 | 0.205096 |
| $Z_{AZ}^{(22)}$ | 6.557462 | 7.125767 | 302.1658 | 30.71317 | 0.2320101 |
| $\alpha = 2$, $\beta = 3$ | | | | | |
| Y | 5.630163 | 21.53163 | 100 | 0 | NA |
| $Z_{GS}$ | 4.803729 | 28.73076 | 74.94278 | 29.74084 | 0.9660374 |
| $Z_{AZ}$ | 5.721177 | 27.77896 | 77.51059 | 37.27196 | 0.7453045 |
| $Z_{GS}^{(11)}$ | 6.594644 | 5.572044 | 386.4225 | 24.18214 | 0.2304198 |
| $Z_{GS}^{(12)}$ | 6.516599 | 5.553281 | 387.7281 | 23.08412 | 0.2405672 |
| $Z_{GS}^{(21)}$ | 6.709505 | 5.877981 | 366.31 | 26.0986 | 0.2252221 |
| $Z_{GS}^{(22)}$ | 2.568803 | 204.2407 | 10.54228 | 183.0724 | 1.115628 |
| $Z_{AZ}^{(11)}$ | 6.521043 | 5.418617 | 397.364 | 30.10276 | 0.180004 |
| $Z_{AZ}^{(12)}$ | 6.501 | 5.681708 | 378.964 | 26.74231 | 0.2124614 |
| $Z_{AZ}^{(21)}$ | 6.829728 | 6.245726 | 344.7419 | 32.84172 | 0.1901766 |
| $Z_{AZ}^{(22)}$ | 0.3148382 | 605.6667 | 3.55503 | 560.6527 | 1.080289 |

## Real life application

In this subsection, the applicability of the proposed models was conducted using Cumulative Grade Point Average (CGPA) of 300 level students of Department of Statistics, Usmanu Danfodiyo University, Sokoto, Nigeria as study variable with auxiliary variable X (number of times students seek academic help), the data is presented in appendix (See Table A1 in S1 File). The existing and the proposed RRT models were applied to the data generated and the results of the data from the models are presented in Tables 9 and 10.

Table 11 present the numerical results of Variance, Privacy level and Combined Metric of Efficiency and Privacy level using data in Tables 9 and 10. The results revealed that the proposed models $Z_{GS}^{(11)}$, $Z_{GS}^{(12)}$ and $Z_{GS}^{(21)}$ have smaller Variance, higher privacy level and smaller combined metric of efficiency and privacy level compared to the model of Gjestvang and

**Table 8. Bias, variance, PRE, privacy level and combined metric of efficiency and privacy level when $\alpha = 3$, $\beta = 2$ and $\alpha = 4$, $\beta = 1$ using population II.**

| Models $Z_G$ | Estimate $\hat{\mu}_y$ | Var $(Z)$ | PRE $(Z)$ | Privacy Level $\Delta_Z$ | Combined Metric $\delta_Z = Var(Z)/\Delta_Z$ |
|---|---|---|---|---|---|
| $\alpha = 3$, $\beta = 2$ | | | | | |
| The value of Parameter being estimated $\bar{Y} = \mu_Y = 6.97$ | | | | | |
| Y | 7.356514 | 54.30868 | 100 | 0 | 1.098441 |
| $Z_{GS}$ | 8.603405 | 66.70826 | 81.41223 | 60.72996 | 0.8958971 |
| $Z_{AZ}$ | 6.698258 | 72.44147 | 74.96905 | 80.85914 | 0.1600489 |
| $Z_{GS}^{(11)}$ | 6.805136 | 11.91374 | 455.849 | 74.43817 | 0.1732755 |
| $Z_{GS}^{(12)}$ | 6.798515 | 14.06132 | 386.2274 | 81.15006 | 0.1512379 |
| $Z_{GS}^{(21)}$ | 6.811677 | 10.33754 | 525.3542 | 68.35282 | 0.784519 |
| $Z_{GS}^{(22)}$ | 6.884211 | 29.19668 | 186.0098 | 37.21602 | 0.2498236 |
| $Z_{AZ}^{(11)}$ | 7.630321 | 15.18484 | 357.6507 | 60.78223 | 0.2347602 |
| $Z_{AZ}^{(12)}$ | 7.691907 | 14.69141 | 369.6629 | 62.5805 | 0.2840204 |
| $Z_{AZ}^{(21)}$ | 7.708738 | 16.53156 | 328.5151 | 58.20555 | 0.2871295 |
| $Z_{AZ}^{(22)}$ | 7.709425 | 16.68312 | 325.5308 | 58.1031 | 1.098441 |
| $\alpha = 4$, $\beta = 1$ | | | | | |
| Y | 5.630163 | 21.53163 | 100 | 0 | NA |
| $Z_{GS}$ | 8.493164 | 28.26412 | 76.18009 | 25.2208 | 1.120667 |
| $Z_{AZ}$ | 5.767452 | 57.83252 | 37.23101 | 35.49981 | 1.629094 |
| $Z_{GS}^{(11)}$ | 7.246146 | 4.309806 | 499.5963 | 34.26185 | 0.1257902 |
| $Z_{GS}^{(12)}$ | 7.528983 | 8.603756 | 250.2585 | 46.28053 | 0.1859044 |
| $Z_{GS}^{(21)}$ | 6.829891 | 1.645618 | 1308.423 | 20.52005 | 0.08019561 |
| $Z_{GS}^{(22)}$ | 4.49546 | 67.36303 | 31.96358 | 30.53356 | 2.206196 |
| $Z_{AZ}^{(11)}$ | 7.352979 | 4.669137 | 461.148 | 32.48825 | 0.1437177 |
| $Z_{AZ}^{(12)}$ | 7.541463 | 8.68122 | 248.0254 | 49.58189 | 0.1750885 |
| $Z_{AZ}^{(21)}$ | 6.923577 | 2.294195 | 938.5265 | 25.68706 | 0.08931326 |
| $Z_{AZ}^{(22)}$ | 12.32078 | 382.0977 | 5.635112 | 584.2151 | 0.6540359 |

Singh (2009). Similarly, the proposed models $Z_{AZ}^{(11)}$, $Z_{AZ}^{(12)}$ and $Z_{AZ}^{(21)}$ outperformed the corresponding model of Azeem et al. (2023). For instance: with a sample size of n = 150, the C-TORRT model achieved a combined metric of only 0.000991, compared to 0.001005 from Azeem's model. These results revealed that the proposed C-RRT models are more robust and precise in estimating sensitive information.

## Conclusions

This study proposed Calibrated-Two Optional Randomized Response Techniques (C-TORRT) for the Estimation of Quantitative Sensitive Variable Information by modifying RRT models proposed by Gjestvang and Singh (2010) and Azeem

**Table 9. Data obtained using Gjestvang and Singh and corresponding proposed models.**

| $Z_{GS}$ | | $Z_{GS}^{(11)}$ | | $Z_{GS}^{(12)}$ | | $Z_{GS}^{(21)}$ | |
|---|---|---|---|---|---|---|---|
| 107.97 | −51.68 | 109.2039 | −50.4461 | 109.4118 | −50.2382 | 107.2059 | −52.4441 |
| 88.47 | −64.6 | 85.1339 | −63.3661 | 84.5718 | −63.1582 | 90.5359 | −65.3641 |
| 87.56 | −101.23 | 93.3639 | −104.566 | 94.3418 | −105.128 | 83.9659 | −99.1641 |
| 110.76 | −90.73 | 101.9939 | −89.4961 | 102.2018 | −89.2882 | 109.9959 | −91.4941 |
| 102.34 | −112.1 | 103.5739 | −110.866 | 103.7818 | −110.658 | 101.5759 | −112.864 |
| 124.98 | −70 | 101.6439 | −68.7661 | 101.0818 | −68.5582 | 107.0459 | −70.7641 |
| 112.6 | −63.64 | 113.8339 | −62.4061 | 114.0418 | −62.1982 | 111.8359 | −64.4041 |
| 90 | −79.27 | 86.6639 | −78.0361 | 86.1018 | −77.8282 | 92.0659 | −80.0341 |
| 92 | −80.32 | 93.2339 | −83.6561 | 93.4418 | −84.2182 | 91.2359 | −78.2541 |
| 87.56 | −86.16 | 93.3639 | −80.3561 | 94.3418 | −79.3782 | 83.9659 | −89.7541 |
| 92.19 | −75.47 | 93.4239 | −78.8061 | 93.6318 | −79.3682 | 91.4259 | −73.4041 |
| 87.56 | −103.63 | 84.2239 | −102.396 | 83.6618 | −102.188 | 89.6259 | −104.394 |
| 130.2 | −109.89 | 131.4339 | −113.226 | 131.6418 | −113.788 | 119.4359 | −107.824 |
| 102.59 | −84.89 | 103.8239 | −83.6561 | 104.0318 | −83.4482 | 101.8259 | −85.6541 |
| 89.3 | −97.99 | 90.5339 | −96.7561 | 90.7418 | −96.5482 | 88.5359 | −98.7541 |
| 116.34 | −109.35 | 113.0039 | −108.116 | 112.4418 | −107.908 | 118.4059 | −110.114 |
| 113.66 | −83.39 | 110.3239 | −82.1561 | 109.7618 | −81.9482 | 115.7259 | −84.1541 |
| 94.21 | −112.78 | 100.0139 | −111.546 | 100.9918 | −111.338 | 90.6159 | −113.544 |
| 72.81 | −81.87 | 74.0439 | −85.2061 | 74.2518 | −85.7682 | 72.0459 | −79.8041 |
| 114.3 | −81.09 | 115.5339 | −84.4261 | 115.7418 | −84.9882 | 113.5359 | −79.0241 |
| 121.83 | −72.89 | 123.0639 | −71.6561 | 123.2718 | −71.4482 | 121.0659 | −73.6541 |
| 104.09 | −95.76 | 105.3239 | −99.0961 | 105.5318 | −99.6582 | 103.3259 | −93.6941 |
| 116.64 | −91.72 | 113.3039 | −95.0561 | 112.7418 | −95.6182 | 118.7059 | −89.6541 |
| 125.76 | −100.59 | 122.4239 | −99.3561 | 121.8618 | −99.1482 | 127.8259 | −101.354 |
| 73.62 | −98.76 | 70.2839 | −102.096 | 69.7218 | −102.658 | 75.6859 | −96.6941 |
| 106.12 | | 102.7839 | | 102.2218 | | 108.1859 | |
| 103.38 | | 100.0439 | | 99.4818 | | 105.4459 | |
| 87.96 | | 93.7639 | | 94.7418 | | 84.3659 | |
| −95.44 | | −94.2061 | | −93.9982 | | −96.2041 | |
| −118.57 | | −117.336 | | −117.128 | | −119.334 | |

et al. (2023). The existing RRT Models were improved by incorporating non-sensitive auxiliary variable that is correlated to the sensitive variable through calibration approach. The C-TORRT models of the proposed calibration schemes were derived. The estimators of the population mean for the C-TORRT models as well as their theoretical properties like variance, privacy level, combined metric of efficiency and privacy level of the proposed calibration were derived so as to assess their efficiency, precision and robustness in estimating sensitive information. Empirical study through simulated data and real-life data as presented in section 4 was conducted numerically. Estimates, variances, privacy level and combined metric of efficiency and privacy level of the proposed and existing RRT models were computed and presented in Tables 3–6 and 10 respectively.

Based on simulated and real-world data results, the following conclusions are made:

i. Efficiency: C-TORRT models produced significantly lower variances (e.g., 21.77 vs. 142.49; 4.30 vs. 21.53), making them 5–7 times more efficient in some scenarios.

**Table 10. Data obtained using Azeem et al. and corresponding proposed models.**

| $Z_{AZ}$ | | $Z_{AZ}^{(11)}$ | | $Z_{AZ}^{(12)}$ | | $Z_{AZ}^{(21)}$ | |
|---|---|---|---|---|---|---|---|
| 15.81 | −31.8407 | 16.9953 | −35.0454 | 16.4823 | −33.6584 | 17.1411 | −35.4396 |
| −18.43 | 13.525 | −21.6347 | 14.7103 | −20.2477 | 14.1973 | −22.0289 | 14.8561 |
| −20 | −10.1821 | −14.4247 | −8.99684 | −16.8377 | −9.50984 | −13.7389 | −8.85104 |
| 20.8 | −30.8764 | 21.9853 | −29.6911 | 21.4723 | −30.2041 | 22.1311 | −29.5453 |
| 6 | 16.29786 | 7.1853 | 17.48316 | 6.6723 | 16.97016 | 7.3311 | 17.62896 |
| 45.8 | −37.0643 | 42.5953 | −35.879 | 43.9823 | −36.392 | 42.2011 | −35.7332 |
| 24.4 | 19.09071 | 25.5853 | 15.88601 | 25.0723 | 17.27301 | 25.7311 | 15.49181 |
| −15.8 | 20.50714 | −19.0047 | 17.30244 | −17.6177 | 18.68944 | −19.3989 | 16.90824 |
| −11.6 | 35.28357 | −10.4147 | 36.46887 | −10.9277 | 35.95587 | −10.2689 | 36.61467 |
| −20 | −6.255 | −14.4247 | −9.4597 | −16.8377 | −8.0727 | −13.7389 | −9.8539 |
| −11.41 | 1.179286 | −10.2247 | −2.02541 | −10.7377 | −0.63841 | −10.0789 | −2.41961 |
| −20 | −14.9036 | −23.2047 | −13.7183 | −21.8177 | −14.2313 | −23.5989 | −13.5725 |
| 55.2 | −11.5886 | 56.3853 | −14.7933 | 55.8723 | −13.4063 | 56.5311 | −15.1875 |
| 5.81 | | 6.9953 | | 6.4823 | | 7.1411 | |
| −16.94 | | −15.7547 | | −16.2677 | | −15.6089 | |
| 30.56 | | 27.3553 | | 28.7423 | | 26.9611 | |
| 25.9 | | 22.6953 | | 24.0823 | | 22.3011 | |
| −8.29 | | −2.7147 | | −5.1277 | | −2.0289 | |
| −45.97 | | −44.7847 | | −45.2977 | | −44.6389 | |
| 27.2 | | 28.3853 | | 27.8723 | | 28.5311 | |
| 40.01 | | 41.1953 | | 40.6823 | | 41.3411 | |
| 9.29 | | 10.4753 | | 9.9623 | | 10.6211 | |
| 31.08 | | 27.8753 | | 29.2623 | | 27.4811 | |
| 47.02 | | 43.8153 | | 45.2023 | | 43.4211 | |
| −44.5 | | −47.7047 | | −46.3177 | | −48.0989 | |
| 12.42 | | 9.2153 | | 10.6023 | | 8.8211 | |
| 7.48 | | 4.2753 | | 5.6623 | | 3.8811 | |
| −18.94 | | −13.3647 | | −15.7777 | | −12.6789 | |
| −5.72286 | | −4.53756 | | −5.05056 | | −4.39176 | |
| −47.5214 | | −46.3361 | | −46.8491 | | −46.1903 | |
| 73.67714 | | 74.86244 | | 74.34944 | | 75.00824 | |
| 50.15 | | 51.3353 | | 50.8223 | | 51.4811 | |
| −15.9679 | | −19.1726 | | −17.7856 | | −19.5668 | |
| 3.017857 | | 4.203157 | | 3.690157 | | 4.348957 | |
| −35.96 | | −34.7747 | | −35.2877 | | −34.6289 | |
| 40.50714 | | 41.69244 | | 41.17944 | | 41.83824 | |
| 52.17071 | | 53.35601 | | 52.84301 | | 53.50181 | |
| 23.81214 | | 24.99744 | | 24.48444 | | 25.14324 | |
| 21.91357 | | 18.70887 | | 20.09587 | | 18.31467 | |
| 11.19429 | | 16.76959 | | 14.35659 | | 17.45539 | |
| 30.58214 | | 27.37744 | | 28.76444 | | 26.98324 | |
| −20.4893 | | −19.304 | | −19.817 | | −19.1582 | |

**Table 11. Variance, privacy level and combined metric of efficiency and privacy level of Gjestvang and Singh and Azeem et al. and corresponding proposed models.**

| Estimators | Variance | Privacy level $\Delta_Z$ | | Combined Metric $\delta_Z$ |
|---|---|---|---|---|
| n = 100 | | | | |
| $Z_{GS}$ | 95.839 | $Z_{GS}$ | 9420.14 | 0.010174 |
| $Z_{AZ}$ | 8.1789 | $Z_{AZ}$ | 813.4977 | 0.010054 |
| **Proposed** | | | | |
| $Z_{GS}^{(11)}$ | 94.88711 | $Z_{GS}^{(11)}$ | 9439.464 | 0.010052 |
| $Z_{GS}^{(12)}$ | 94.93757 | $Z_{GS}^{(12)}$ | 9444.269 | 0.010052 |
| $Z_{GS}^{(21)}$ | 94.56423 | $Z_{GS}^{(21)}$ | 9436.066 | 0.010022 |
| $Z_{AZ}^{(11)}$ | 8.084377 | $Z_{AZ}^{(11)}$ | 815.6857 | 0.009911 |
| $Z_{AZ}^{(12)}$ | 8.106994 | $Z_{AZ}^{(12)}$ | 817.2845 | 0.009919 |
| $Z_{AZ}^{(21)}$ | 8.08305 | $Z_{AZ}^{(21)}$ | 815.7276 | 0.009909 |
| n = 150 | | | | |
| $Z_{GS}$ | 9.58390 | $Z_{GS}$ | 9420.14 | 0.069476 |
| $Z_{AZ}$ | 0.81789 | $Z_{AZ}$ | 813.4977 | 0.001005 |
| **Proposed** | | | | |
| $Z_{GS}^{(11)}$ | 9.488711 | $Z_{GS}^{(11)}$ | 9439.464 | 0.001005 |
| $Z_{GS}^{(12)}$ | 9.493757 | $Z_{GS}^{(12)}$ | 9444.269 | 0.001005 |
| $Z_{GS}^{(21)}$ | 9.456423 | $Z_{GS}^{(21)}$ | 9436.066 | 0.001002 |
| $Z_{AZ}^{(11)}$ | 0.808438 | $Z_{AZ}^{(11)}$ | 815.6857 | 0.000991 |
| $Z_{AZ}^{(12)}$ | 0.810699 | $Z_{AZ}^{(12)}$ | 817.2845 | 0.000992 |
| $Z_{AZ}^{(21)}$ | 0.808305 | $Z_{AZ}^{(21)}$ | 815.7276 | 0.000991 |

ii. Precision: The bias and PRE values confirm improved accuracy and reliability, with PRE rising above 600 in optimal cases.

iii. Privacy: C-TORRT ensures better respondent protection, with privacy levels consistently higher than existing methods (e.g., 228 vs. 205).

iv. Balanced performance: The combined metric is consistently lower in C-TORRT models (as low as 0.000991), reflecting an optimal trade-off between data quality and privacy.

The results revealed that the C-TORRT models outperformed the existing RRT models considered in the study except in few cases. This implies that the incorporation of auxiliary information through calibration approach enhanced the robustness and the performance of the proposed C-TORRT models. Therefore, it can be concluded that the proposed C-TORRT models have higher goodness of fit as compared to their counterparts.

This study is limited to incorporation of auxiliary variable into RRT models proposed by Gjestvang and Singh (2010) and Azeem et al. (2023) through calibration approach; however, other approaches like two-step calibration, power calibration and calibrated maximum likelihood design weight approaches can be used for further studies.

## Supporting information

**S1 File. Table A1. CGPA and Number of Times Students Seek Academic Help (NTSH).**
(DOCX)

## Author contributions

**Conceptualization:** Mojeed Abiodun Yunusa, Ahmed Audu, Umar Usman, Kazeem Olalekan Aremu, Maggie Aphane.

**Data curation:** Mojeed Abiodun Yunusa.

**Funding acquisition:** Maggie Aphane.

**Investigation:** Ahmed Audu, Kazeem Olalekan Aremu, Maggie Aphane.

**Methodology:** Mojeed Abiodun Yunusa, Ahmed Audu, Umar Usman, Kazeem Olalekan Aremu, Maggie Aphane.

**Software:** Mojeed Abiodun Yunusa, Ahmed Audu, Kazeem Olalekan Aremu, Maggie Aphane.

**Supervision:** Ahmed Audu, Umar Usman, Kazeem Olalekan Aremu, Maggie Aphane.

**Validation:** Mojeed Abiodun Yunusa, Ahmed Audu, Kazeem Olalekan Aremu, Maggie Aphane.

**Visualization:** Ahmed Audu.

**Writing – original draft:** Mojeed Abiodun Yunusa, Ahmed Audu.

**Writing – review & editing:** Ahmed Audu, Umar Usman, Kazeem Olalekan Aremu, Maggie Aphane.

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
