## [Decision Letter · Decision Letter 0]

16 May 2025

Dear Dr. Audu,

Thank you for submitting your manuscript to PLOS ONE. After careful consideration, we feel that it has merit but does not fully meet PLOS ONE’s publication criteria as it currently stands. Therefore, we invite you to submit a revised version of the manuscript that addresses the points raised during the review process.

Both reviewers recommended revision. Please work on thoroughly addressing reviewers' comments and get back to PLOS One by the designated deadline. 

We look forward to receiving your revised manuscript.

Kind regards,

Chenfeng Xiong

Academic Editor

PLOS ONE

2. We note that your Data Availability Statement is currently as follows: All relevant data are within the manuscript and in Supporting Information files.

Reviewers' comments:

Reviewer's Responses to Questions

**Comments to the Author**

1. Is the manuscript technically sound, and do the data support the conclusions?

Reviewer #1: Yes

Reviewer #2: Partly

2. Has the statistical analysis been performed appropriately and rigorously?

Reviewer #1: Yes

Reviewer #2: Yes

3. Have the authors made all data underlying the findings in their manuscript fully available?

Reviewer #1: Yes

Reviewer #2: Yes

4. Is the manuscript presented in an intelligible fashion and written in standard English?

Reviewer #1: Yes

Reviewer #2: No

Reviewer #1: 1. Authors are suggested to rewrite (clear and structured) the Abstract which will improve the accessibility of the paper and help readers grasp its core contributions quickly.

2. The authors should carefully revise the manuscript to eliminate any unnecessary repetition.

3.Provide practical examples of applications.

4. Recheck the manuscript for various grammatical and typo errors.

Reviewer #2: The presented work entitled " Calibrated-Two Optional Randomized Response Techniques (C-TORRT) for the

Estimation of Quantitative Sensitive Variable Information" need major revision but the current form is not acceptable.

1. Abstract should rewrite as it should be indicated the main findings (with some values).

2. Abstract should not contain any references.

3. Reference citation is irregular. It should be in sequence.

4. The first paragraph of introduction part is without any reference.

5. Equations are not cited properly. Give the refence for each equation.

6. The entire work shows without single results (in terms of graphical representation).

7. In some of the sentence, it is difficult to understand the meaning of text.

8. There is no comparative study with existing work.

9. The data and results are not sufficient. Author must add more results otherwise work will not recommend for publication.

10. Result explanation is not sufficient and justified. Give proper reason for all the results and justify.

11. Conclusion should have some resulting details with numerical data.

**Do you want your identity to be public for this peer review?** For information about this choice, including consent withdrawal, please see our Privacy Policy

Reviewer #1: No

Reviewer #2: No

---

## [Author Response · Author response to Decision Letter 1]

30 Jun 2025

PONE-D-25-03382

Calibrated-Two Optional Randomized Response Techniques (C-TORRT) for the Estimation of Quantitative Sensitive Variable Information

Reviewer #1:

S/N REVIEWER’S COMMENTS AUTHORS’ RESPONSES

1 Authors are suggested to rewrite (clear and structured) the Abstract which will improve the accessibility of the paper and help readers grasp its core contributions quickly. DONE

2 The authors should carefully revise the manuscript to eliminate any unnecessary repetition. DONE

3 Provide practical examples of applications. DONE. SEE PAGE 26-28

4 Recheck the manuscript for various grammatical and typo errors. DONE

Reviewer #2:

S/N REVIEWER’S COMMENTS AUTHORS’ RESPONSES

1 Abstract should rewrite as it should be indicated the main findings (with some values). DONE

2 Abstract should not contain any references DONE

3 Reference citation is irregular. It should be in sequence. DONE

4 The first paragraph of introduction part is without any reference. DONE

5 Equations are not cited properly. Give the reference for each equation. CORRECTIONS ON EQUATIONS HAVE BEEN DONE

6 The entire work shows without single results (in terms of graphical representation). SINCE GRAPHS OF THE RESULTS WILL NOT GIVE DIFFERENT RESULTS, THEY ARE EXEMPTED TO MINIMIZE THE PAGE NUMBER OF THE MANUSCRIPT

7 . In some of the sentence, it is difficult to understand the meaning of text. CORRECTED

8 There is no comparative study with existing work. THE PROPOSED CALIBRATED RRT MODELS WERE COMPARED WITH THAT OF Gjestvang and Singh (2009) AND Azeem et al (2023) . See tables 2a-7

9 The data and results are not sufficient. Author must add more results otherwise work will not recommend for publication DONE. SEE TABLE 5-7

10 Result explanation is not sufficient and justified. Give proper reason for all the results and justify. DONE

11 Conclusion should have some resulting details with numerical data. DONE

---

## [Decision Letter · Decision Letter 1]

4 Dec 2025

Calibrated-Two Optional Randomized Response Techniques (C-TORRT) for the Estimation of Quantitative Sensitive Variable Information

PONE-D-25-03382R1

Dear Dr. Audu,

We’re pleased to inform you that your manuscript has been judged scientifically suitable for publication and will be formally accepted for publication once it meets all outstanding technical requirements.

Kind regards,

Chenfeng Xiong

Academic Editor

PLOS One

Additional Editor Comments (optional):

Reviewers' comments:

Reviewer's Responses to Questions

**Comments to the Author**

Reviewer #1: All comments have been addressed

Reviewer #2: All comments have been addressed

2. Is the manuscript technically sound, and do the data support the conclusions?

Reviewer #1: Yes

Reviewer #2: Yes

3. Has the statistical analysis been performed appropriately and rigorously?

Reviewer #1: Yes

Reviewer #2: Yes

4. Have the authors made all data underlying the findings in their manuscript fully available?

Reviewer #1: Yes

Reviewer #2: Yes

5. Is the manuscript presented in an intelligible fashion and written in standard English?

Reviewer #1: Yes

Reviewer #2: Yes

Reviewer #1: In the revised manuscript, authors have incorporated the suggestions, and the paper can be accepted in its current form.

Reviewer #2: No more comments as the author has incorporated all the comments in the revised manuscript. now the manuscript may be accepted.

**Do you want your identity to be public for this peer review?** For information about this choice, including consent withdrawal, please see our Privacy Policy

Reviewer #1: **Yes: ** Chandraketu Singh

Reviewer #2: No

---

## [Editor Report · Acceptance letter]

PONE-D-25-03382R1

PLOS One

Dear Dr. Audu,

I'm pleased to inform you that your manuscript has been deemed suitable for publication in PLOS One. Congratulations! Your manuscript is now being handed over to our production team.

Kind regards,

on behalf of

Dr. Chenfeng Xiong

Academic Editor

PLOS One